# The AERosol and TRACe gas Collector (AERTRACC): an online measurement controlled sampler for source-resolved emission analysis

Julia Pikmann[1], Lasse Moormann[1], Frank Drewnick[1], Stephan Borrmann[1,2]

[1]Particle Chemistry Department, Max Planck Institute for Chemistry, Mainz, 55128, Germany
[2]Institute for Atmospheric Physics, Johannes Gutenberg University Mainz, Mainz, 55128, Germany

*Correspondence to*: Frank Drewnick (frank.drewnick@mpic.de)

**Abstract.**

Probing sources of atmospheric pollution in complex environments often leads to the measurement and sampling of a mixture of different aerosol types due to fluctuations of the emissions or the atmospheric transport situation. Here, we present the AERosol and TRACe gas Collector (AERTRACC), a system for sampling various aerosol types independently on separate sampling media, controlled by parallel online measurements of particle, trace gas, and meteorological variables, like particle number or mass concentration, particle composition, trace gas concentration as well as wind direction and speed. AERTRACC

is incorporated into our mobile laboratory (MoLa) which houses online instruments measuring various physical and chemical aerosol properties as well as trace gas concentrations. Based on preparatory online measurements with the whole MoLa setup, suitable parameters measured by these instruments are used to define individual sampling conditions for each targeted aerosol type using a dedicated software interface. Through evaluation of continuously online measured data with regard to the sampling conditions, the sampler automatically switches between sampling and non-sampling for each of up to four samples, which can

be collected in parallel. Particle and gas phase of each aerosol type, e.g. source emissions and background, are sampled onto separate filters  with $PM_1$ and $PM_{10}$ cutoffs, respectively, and thermal desorption tubes, respectively. Information on chemical compounds in the sampled aerosol is accomplished by thermal desorption chemical ionization mass spectrometry (TD-CIMS) as analysis method. The design, operation, and characterization of the sampler are presented. For in-field validation, wood-fired pizza oven emissions were sampled as targeted emissions separately from ambient background. Results show that the

combination of well-chosen sampling conditions allows more efficient and effective separation of source-related aerosols from the background, as seen by increases of particle number and mass concentration and concentration of organic aerosol types, with minimized loss of sampling time, compared to alternative sampling strategies.

## 1 Introduction

Atmospheric aerosol changes radiative forcing, alters cloud formation and precipitation, and affects human health. Various
chemical and physical processes lead to changes of the aerosol properties, like the particle size and composition (Fuzzi et al., 2015; Johnston und Kerecman, 2019; Shrivastava et al., 2017). Still the impact of these effects on climate and health are not

sufficiently well understood as aerosol sources, composition, properties, and transformations are poorly characterized (Parshintsev und Hyötyläinen, 2015).

Atmospheric aerosol can originate from diverse sources, natural as well as anthropogenic ones. Primary particles can be related
to anthropogenic sources like combustion processes of fossil fuel and biomass as well as natural sources emitting e.g. sea salt and dust. Furthermore, secondary aerosol forms through gas-to-particle conversion by oxidation processes in the atmosphere (Celik et al., 2020; Fuzzi et al., 2015; Gordon et al., 2017; Struckmeier et al., 2016). Depending on the surroundings, different types of emissions and the background aerosol can blend into complex mixtures, complicating the identification of the contribution by the original emissions sources.

Atmospheric aerosol are generally classified into two major chemical fractions, the inorganic one with substances like ammonium, nitrate, sulfate, metal oxides, mineral dust, and sea salt, while the organic aerosol, the other fraction, constitutes the more complex part (Fuzzi et al., 2015). Especially fine particulate matter, which has a relevant effect on climate and health, contains usually a large organic fraction (Zheng et al., 2020). These particles consist of many individual components but only a small fraction of them are identified by state-of-the-art instruments (Fuzzi et al., 2015; Johnston und Kerecman, 2019; Zhou
et al., 2020). The analysis and identification of these organic components is necessary for better understanding of chemical processes, transport, sources, and particle formation in the atmosphere. This knowledge is crucial to improve existing models and facilitate prediction of climate effects (Johnston und Kerecman, 2019; Zhou et al., 2020).

Characterization of organic aerosol is demanding due to the broad variety of species and therefore numerous techniques for aerosol analysis have been developed (Forbes, 2020; Johnston und Kerecman, 2019). Techniques for analysis and
characterization of aerosols are classified into two main categories, online and offline techniques. Offline measuring techniques frequently provide detailed information about different aerosol properties based on separate sampling and analysis (Parshintsev und Hyötyläinen, 2015). For chemical analysis, this approach offers the possibility to use all available analysis techniques to get detailed information at the expense of low time and particle size resolution (Hallquist et al., 2009; Heard, 2006). A broad variety of techniques are available for chemical analysis. Techniques like ICP-MS (inductively coupled plasma mass
spectrometry) and XRF (x-ray fluorescence) provide information about the elemental composition of the sample (Bhowmik et al., 2022; Ebert et al., 2016), while FTIR (Fourier-transform infrared spectroscopy) and NMR (nuclear magnetic resonance spectroscopy) are used to determine organic functional groups in aerosols (Faber et al., 2017; Gilardoni, 2017).Techniques with separation prior to detection are applied for identification of individual species. Widely used for this purpose are GC-MS (gas chromatography mass spectrometry) and HPLC-MS (high performance liquid chromatography mass spectrometry);
however they are typically only able to identify a relatively small fraction of the whole organic aerosol (Forbes, 2020). Single particle techniques like SIMS (secondary ion mass spectrometry) and SEM (scanning electron microscope) provide information about the elemental composition and its distribution as well as information about the particle morphology (Bai et al., 2018; Laskin et al., 2018).

Online and semi-online techniques are used to obtain data with high time resolution. With these techniques, samples are
analyzed continuously or semi-continuously without the need of additional a-posteriori laboratory work as for offline

techniques. One of the most widely used methods for aerosol online analysis is aerosol mass spectrometry (AMS) measuring the single particle or particle ensemble chemical composition of submicron particles. While offering real-time data due to short acquisition intervals it lacks detailed chemical information, lost through fragmentation during vaporization and ionization (Canagaratna et al., 2007). Consequently, identification of individual organic components is rarely possible (Hallquist et al.,

2009). A semi-continuous online bulk analysis can be performed with the thermal-optical EC/OC analyzer measuring the hourly concentrations of elemental carbon (EC) and organic carbon (OC) (Zhou et al., 2015). Other semi-continuous systems like PILS (particle into liquid sampler) and MARGA (monitor for aerosols and gases in ambient air) sample the water-soluble aerosol fraction followed by subsequent analysis with e.g. ion chromatography (Stavroulas et al., 2019; Zhou et al., 2015). More comprehensive analysis is achieved with TAG (thermal desorption aerosol gas chromatography) (Williams et al., 2006)

and FIGAERO-CIMS (filter inlet for gas and aerosols chemical ionization mass spectrometry) (Lopez-Hilfiker et al., 2014), which sample aerosol for several tens of minutes and analyze the samples after automated thermal desorption. These semi-continuous techniques offer rather detailed information on the organic aerosol fraction due to low fragmentation. However, with time resolutions of tens of minutes up to an hour, characterization of transient emissions or disentanglement of aerosol blends in environments affected by several sources is not feasible

A few instruments with high time resolution in the order of seconds, sufficient for the analysis of transient aerosol occurrences, combined with detailed analysis were developed in recent years, such as the EESI-ToF (electrospray ionization time-of-flight mass spectrometer) (Lopez-Hilfiker et al., 2019; Pagonis et al., 2021) and the CHARON-PTR-MS (chemical analysis of aerosol online proton-transfer-reaction mass spectrometer) (Eichler et al., 2015; Piel et al., 2019).

To comprehensively analyze and characterize individual sources in complex environments like cities or industrial areas, where

fluctuating meteorological and atmospheric transport conditions result in mixing of emissions from different sources, or transient source emissions like from ships, aircrafts or short-term processes, identification of individual species on short time scales is necessary. Offline and semi-online methods offering highly resolved speciation data do not provide the required temporal resolution and high-time resolution online methods typically do not provide in-depth chemical analysis capability. Therefore, we developed the AERosol and TRACe gas Collector (AERTRACC), which combines the advantages of both

approaches. AERTRACC collects samples of different aerosol types for subsequent in-depth analysis on separate sampling media which can be quickly and simply exchanged. Separation of aerosol types is hereby achieved by controlling the sampling process with high-time resolution online measurements. AERTRACC is integrated in our mobile aerosol research laboratory (MoLa), a vehicle equipped with online measuring instruments (Drewnick et al., 2012), serving as control unit for the sampler via a tailor-made software interface. There the user can define sampling conditions based on measured parameters like particle

number concentration or wind direction to separately collect the different aerosol types. While online instruments for in-depth chemical analysis with high temporal resolution are limited to the respective analysis methods, the AERTRACC sampler enables the use of the full potential of analytical chemistry and microscopic analysis for the investigation of such aerosols beyond these specific approaches. For this work, TD-HR-ToF-CIMS (thermal desorption high resolution time-of-flight chemical ionization mass spectrometry) was chosen as analysis method offering high resolution mass spectra combined with

high sensitivity and low sample fragmentation as well as minimized sample preparation effort (Aljawhary et al., 2013; Mercier et al., 2012; Yatavelli et al., 2012). Here, we present the design and characteristics of AERTRACC and demonstrate its capabilities in a field experiment, probing the emissions of a pizza oven in a semi-urban environment.

## 2 Design and operation of the AERTRACC sampling system

### 2.1 The mobile aerosol research laboratory (MoLa)

The mobile laboratory MoLa houses the newly developed AERTRACC sampling system and serves as data-providing basis for its control unit. MoLa is designed for mobile and stationary measurements of ambient air composition and is mainly used for characterization of source specific emissions (Drewnick et al., 2012; Fachinger et al., 2021). A variety of online instruments measures different aerosol and meteorological properties providing high time resolution data of seconds until one-minute averaging intervals. This includes physical particle properties, e.g. particle number size distributions, as well as chemical characterization like the non-refractory chemical composition of submicron particles, and trace gas concentrations of various gases as $NO_x$, $O_3$, and $CO_2$. An overview of the MoLa instruments and measured variables, which are used to control the AERTRACC system, is provided in Table 1; for further description see Drewnick et al. (2012). Stationary measurements can be performed with the sampling inlet at different heights (3-10 m above ground level) using an inlet setup on MoLa's roof.

Table 1: MoLa instruments used for control of the AERTRACC sampler.

| Instrument | Measured variables | Particle diameter range | Time resolution |
|---|---|---|---|
| Aethalometer[a] | Black and brown carbon mass concentration | < 1.0 µm | 1 s |
| PAS[b] | Polyaromatic hydrocarbon mass concentration on particle surface | 10 nm - 1 µm | 12 s |
| EDM[c] | $PM_1$, $PM_{2.5}$, $PM_{10}$ mass concentration based on optical measured size distribution | 0.25 - 10 µm | 6 s |
| CPC[d] | Particle number concentration | 5 nm - 3 µm | 1 s |
| OPC[e] | Particle size distribution based on optical diameter | 0.25 - 32 µm | 6 s |
| Airpointer[f] | Mixing ratio of CO, $SO_2$, $O_3$. $NO_x$ | - | 4 s |
| $NO_2$/NO/$NO_x$ Monitor[g] | Mixing ratio of $NO_2$, NO, $NO_x$ | - | 5 s |
| LICOR[h] | Mixing ratio of $CO_2$. $H_2O$ | - | 1 s |
| Meteorological Station[i] | Wind direction, wind speed, relative humidity, temperature, rain intensity, pressure | - | 1 s |

| GPS[j] | Location | - | 1s |
| HR-ToF-AMS[k] | Size-dependent non-refractory chemical composition | 40 nm - 1 µm | 15 s |

[a]Magee Scientific Aethalometer® Model AE33, Magee Scientific, USA. [b]Photoelectric Aerosol Sensor PAS2000, EcoChem Analytics, USA. [c]Environmental Dust Monitor EDM180, Grimm Aerosoltechnik, Germany. [d]Condensation Particle Counter Model 3786, TSI, Inc., USA. [e]Optical Particle Counter Model 1.109, Grimm Aerosoltechnik, Germany. [f]AirPointer, Recordum Messtechnik GmbH, Austria. [g]NO2/NO/NOx Monitor Model 405 nm, 2B Technologies, Inc., USA. [h]LI840, LI-COR, Inc., USA. [i]WXT520, Vaisala, Finland. [j]Navilock NL-8022MU, Navilock, Germany. [k]High-resolution Time-of-Flight Aerosol Mass Spectrometer, Aerodyne Research, Inc., USA, (currently not used for AERTRACC control, but might be implemented for future studies, was used for theoretical sampling scenarios).

## 2.2 Setup of the AERTRACC sampling system

AERTRACC is designed to sample different aerosol types separately on individual sample carriers. The system is incorporated in MoLa with its own inlet and a flow path designed for minimal particle losses, minimizing non-vertical tubes and bends. With four available sampling paths up to four different aerosol types can be sampled separately. It is possible to sample particles with two different size cuts on quartz fiber or PTFE filters as well as volatile compounds onto thermal desorption tubes (TDT) filled with adsorbent material (further details in Sect. 2.4). A control software for the AERTRACC sampler was programmed to accomplish separate sampling of different aerosol types based on the MoLa online data (see Sect. 2.3).

A schematic overview and a photograph of the sampling system installed in MoLa are shown in Fig. 1. The AERTRACC sampler has its own inlet line (ID = 48 mm), equipped with a $PM_{10}$ inlet head (Digitel, Switzerland, inlet flow rate 30 L min$^{-1}$) for sampling nominal $PM_{10}$, which is mounted on the roof of MoLa. The inlet is located 0.5 m apart from the MoLa online instrument inlet and their heights are adjusted to each other to assure sampling of the same aerosol.

Inside MoLa the inlet tube is split into two main paths, which are both split again, in total into four sampling paths. Main path 1 (see Fig. 1b) contains a $PM_1$ cyclone (URG, USA, flow rate 16.7 L min$^{-1}$).and is connected to main path 2 with a cross tube downstream of the $PM_1$ cyclone. With two ball valves, one installed in main path 2 and the other one in the cross tube between the main paths, the user can sample in two different sampling modes. Either two sampling paths are used for $PM_{10}$ and the other two for $PM_1$ (Fig. 1b; cross tube not used) or all four sampling paths are used for $PM_1$ sampling (Fig. 1c; cross tube used to feed also main path 2 through the cyclone).

Each of the four sampling paths contains a custom-made filter holder made of gold-coated aluminum for filters of 25 mm diameter and a TDT. The sampling area on the filters equals the thermal desorption area for the subsequent analysis. The operation flow rate for filter sampling is limited to 4.2 or 7.5 L min$^{-1}$ (¼ of 16.7 or 30 L min$^{-1}$) due to the required flow rates for the $PM_1$ cyclone or the $PM_{10}$ inlet, depending on the chosen sampling mode (see above).The sampling line splits again behind each filter holder into a path with TDT and a TDT bypass path. This split is necessary, as the flow rate through the TDT has to be smaller (typically limited to 0.2 L min$^{-1}$) than the one through the filter to avoid a loss of the retention volume for the gaseous species. The described active sampling paths are shown in Fig. 1b as green paths. The flows through the filter

holders are the sum of the flows through the respective TDT and TDT bypass lines. Simple and quick change of filter holders and TDTs is achieved with Ultra-Torr vacuum fittings (Swagelok Company, USA) before and behind each device.

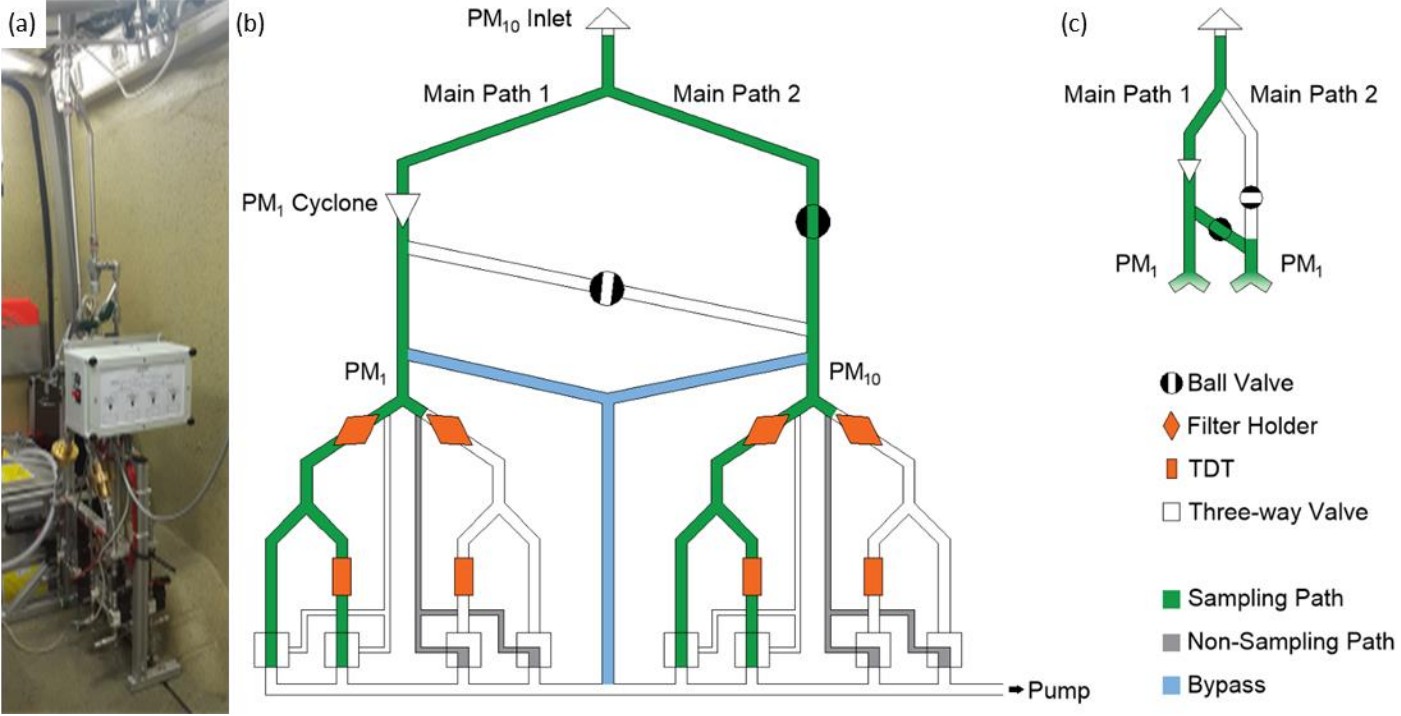

**Figure 1: Photo (a) and scheme (b+c) of the AERTRACC sampler with active flow paths marked in green for sampling flows, in grey for non-sampling flows, and in blue for bypass flows (b: sampling mode $PM_1$+$PM_{10}$, c: sampling mode $PM_1$ only). Needle and dosing valves are located in each path directly upstream of the pump and are not shown in the scheme.**

To assure a permanent air flow through the whole system, independent whether a certain sample line is active or not, a non-sampling path around the sampling media is added in parallel to each sampling line (grey paths in Fig. 1b are the active non-sampling paths). Further, for non-sampling conditions, diffusion of volatiles from one sampling path to another is avoided as the volatiles would have to diffuse a short distance upstream the flow persisting through the non-sampling path. The flow through the sampling system is switched between sampling and non-sampling path using magnetic three-way valves (SMC, VT307, Japan) and maintained by a rotary vane pump (V-VTE 10, Gardner Denver, Inc., USA). This permanent air flow through the system keeps the cut-offs of the size selectors and the transport losses constant and allows the targeted aerosol to be sampled almost immediately as soon as the respective three-way valve is switched when the evaluation of the online data shows that the sampling conditions are fulfilled. The adjustment of flow rates for the sampling paths is achieved with precision dosing valves (HF-1300-SS-L-1/4-S, Hamlet, Germany) for the TDT flow rates and with needle valves (Nupro SS-4HS V51, Swagelok, USA) for the additional flow through the filters before each experiment. No change of flow rates was observed during test measurements. Replacing the needle valves by mass flow controllers for future studies is planned to ensure constant flow rates and to simplify flow settings.

Independent of the individual sample line flow rates, an additional bypass line is split from each main path (blue paths in Fig. 1b) to adjust the flows through the two main paths to match the specified flow rates through the inlet head and the cyclone. These bypass lines are directly connected to the pump via additional needle valves. The sampling line and bypass tubing are made of stainless steel with tube diameters of 1/2" upstream the filter holders and 1/4" after the filter holders.

The AERTRACC electronics including the control of the magnetic valves via a custom-made relay card and relays is housed in an electronic box attached to the sampler (white box in Fig. 1a). The front of the box contains an LED status display showing which sampling path is active. The relay card is connected via RS232 to the MoLa data acquisition computer, which collects the online instruments data.

## 2.3 Control software and sampler operation

The AERTRACC control software (ACS) is the interface between the MoLa online measurements and the sampling system and is integrated into the MoLa data acquisition software for simple and direct access to the data. It was developed in Igor Pro (Version 6.3, WaveMetrics, Inc., USA) and is available from the authors upon request. In the ACS, the user defines criteria for sampling up to four different aerosol types separately, based on measured MoLa online data. The software continuously evaluates the incoming online data whether the criteria for sampling are fulfilled and controls the flow through the individual sampling paths accordingly.A graphical user interface (Fig. 2) was programmed for effective and user-friendly operation where the user selects the sampling conditions for the targeted aerosol types and obtains real-time information on the sampling process, such as the accumulated sampling time and estimated collected mass on the filters. In the upper part of the main ACS window, the user chooses the operation and sampling mode. The lower part is divided into four boxes, one for each sampling path, where the user can set sampling conditions individually for each path.

Two sampling modes are available, $PM_1$ and $PM_1+PM_{10}$. For the $PM_1+PM_{10}$ sampling mode, the same sampling conditions are used for each $PM_1$ / $PM_{10}$ sampling path pair. The user can choose between two operation modes. The sampler can either be operated in *automatic mode* with user defined sampling conditions (Fig. 2a), which are based on variables, measured by the MoLa online instruments, or in *manual mode* (Fig. S4), where the user can directly start and stop sampling with the additional possibility to pre-select the collection time or collected mass on the filters. The total collected mass on the filters is calculated based on the EDM online mass concentration data, measured during the actual sampling intervals, and the respective filter flow rate.

In the *automatic mode* the user defines individual sampling conditions for each sampling path (Fig. 2a). Each sampling condition consists of up to four criteria, which can be logically combined using the Boolean operators AND, NOT, and OR. Individual criteria are fulfilled if the value of the selected parameter, e.g. a particle or trace gas concentration, but also time, GPS location, meteorological condition, or total collected mass, is between the user-selected minimum and maximum values. This allows complex definitions of sampling conditions for each of the targeted aerosol types. A possible scenario, based on recent MoLa measurements (Fachinger et al., 2021), could be measuring with MoLa at a place where traffic and biomass burning emissions can be measured depending on the wind direction.Both types of emissions could be sampled separately

using suitable sampling conditions. For the biomass burning aerosol the sampling condition could be "suitable wind direction range AND high black carbon concentration AND high $PM_1$ concentration"; while for the traffic aerosol the sampling condition could be "suitable wind direction range AND high particle number concentration AND NOT high $PM_1$ concentration". For background aerosol sampling the mentioned variables should be accordingly set to low concentrations and the remaining wind direction sections.

During measurements when air masses containing different aerosol types reach the inlet, the sampler switches automatically between the according sampling paths based on the evaluation of the sampling conditions. The evaluation is performed each second based on the highest available time resolution of the instruments, hence the valves can be switched on a 1s-base as well. While frequent switching of the valves introduces frequent flow and pressure disruptions in the sampler, these are not expected to produce enhanced sampling artefacts by e.g. re-volatilization of material from the tube surface or the filters,

compared to less frequent switching scenarios. Therefore, switching between different sampling paths typically occurs multiple times within an experiment of hours of duration, which is in contrast to conventional continuous sampling. Although the AERTRACC is primarily designed for stationary measurements, it is also possible to sample during mobile measurements if the air mass segments are large enough to differentiate between them on a few seconds time scale.The *flowrate* sub-window contains information on the flow setup of the AERTRACC sampler (Fig. 2b). Here, the user enters the flow rates, which are

adjusted with the individual needle valves. The graphical user interface automatically provides the combined flow rates at critical devices, such as the inlet cyclone, and thus supports the correct selection of the individual flow rates in order to match their required flow conditions. Furthermore, in this window the MoLa inlet height is entered. This information is used to select the correct delay times between registration of the sampling status, i.e. sampling or non-sampling, and the activation or de-activation of flows through the individual sampling paths (see Sect. 3.2).

When the sampling path is activated, the software continuously compares the chosen sampling conditions with the actual measured online data. For visual support a colored indicator shows for each sampling path whether sampling (green) or no sampling (red) takes place or the sampling path is inactive (grey). Depending on whether the sampling conditions for a certain sampling path are fulfilled, the respective three-way valves are switched accordingly between sampling path and non-sampling path via the relay card.Two displays in the ACS for each sampling path show the current accumulated collection time and

sampled aerosol mass. A data logger automatically keeps track of all activities performed by the user on the interface and of all sampling periods, which are logged with the time stamp, type of activity and respective sampling conditions.

The chemical analysis of aerosol samples (like e.g. when using FIGAERO-CIMS measurements of organic compounds) typically requires sampled mass in the order of 1 µg in order to exceed instrumental detection limits, depending on the specific analysis method. In urban conditions with organic mass concentrations of 5-10 µg m$^{-3}$ (Chen et al., 2022), with a sample flow

rate of 7.5 L min$^{-1}$ and with approximately 10 % of the time sampling source emissions (like in our validation experiment, see Section 4), a total sampling time in the order of 1-2 hours would be needed to collect enough material for analysis. Therefore, the probed source must emit over sufficiently long times to allow a successful chemical characterization of their emissions. Higher emission concentrations, more stable transport conditions, and lower detection limits of the applied analysis method

can reduce sampling times significantly. Especially when using microscopic and single particle techniques, which might need extremely low amounts of sample, sampling times could be reduced further and also single transient emission events might provide sufficient material for successful analysis.

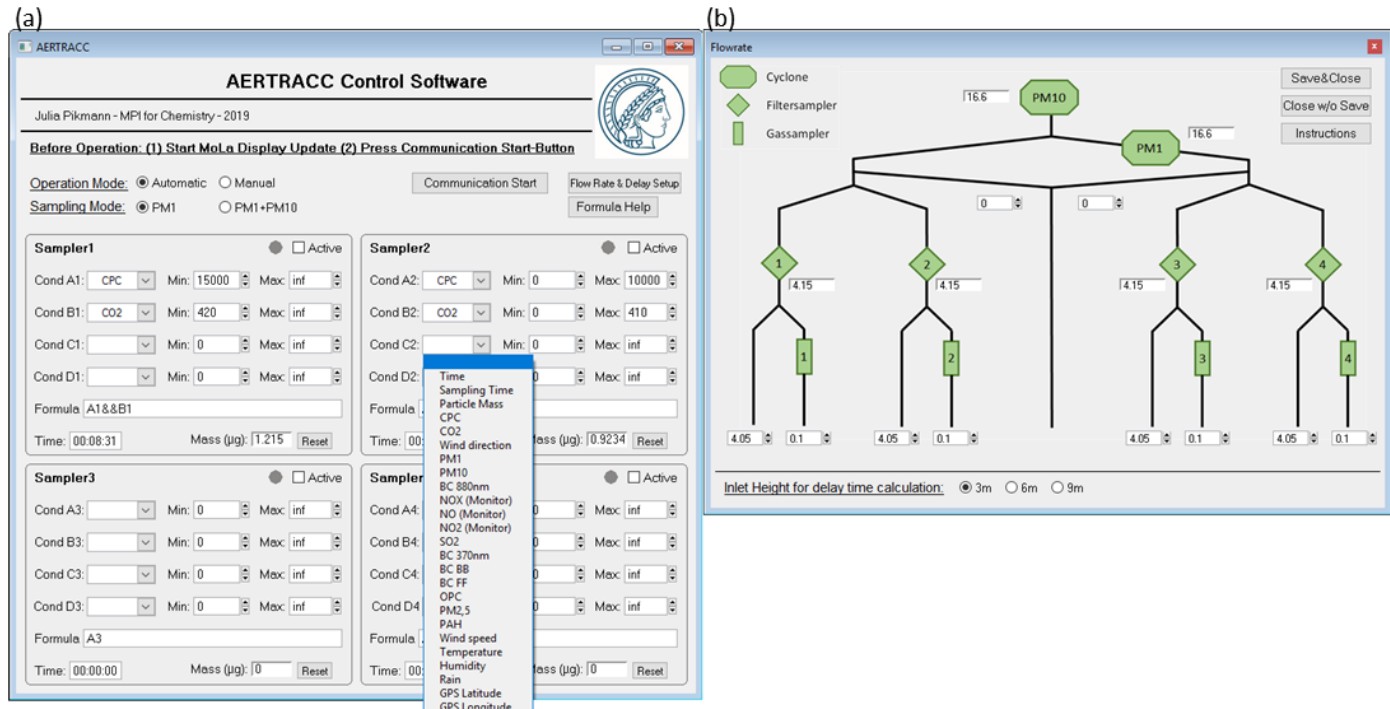

Figure 2: User interface of the AERTRACC software with main window (a) and flowrate sub-window (b).

## 2.4 Sampling media

Generally, the sampling media used are dependent on the subsequent offline analysis method. The choice of the sampling media for this study was based on the selection of thermal desorption as sample introduction method for the subsequent analysis using TD-CIMS, which reduces the chances of potential contamination through sample preparation. For gas phase sampling, TDTs were used, made of stainless steel (1/4" OD, 89 mm length) and packed with Tenax TA (MS Wil, Netherlands) and Carbograph 5TD (Markes International Ltd., United Kingdom), each 150 mg. Together, these adsorbents are applicable to compounds with a broad volatility range (mainly $C_4 - C_{32}$) to investigate different kinds of emissions. They were also chosen as they are hydrophobic, inert and temperature stable up to 350 °C, necessary for the high temperature during thermal desorption (Dettmer und Engewald, 2002, 2003; Harper, 2000; Woolfenden, 2010). TDTs were conditioned in a TC-20 conditioner (Markes International Ltd., United Kingdom) at 300 °C for 4 hours with nitrogen (purity 99.9999%, 0.09 L min⁻¹) before sampling.

For particle sampling, PTFE filters with 25 mm diameter (Type 11803, Sartorius, Germany) were used, which were pre-baked at 200 °C under vacuum (50 hPa) for 24 h before sampling.

Typical sampling flow rates are usually between 1 and 8 L min$^{-1}$ for the filter samples, with mass loadings not exceeding 2 µg to avoid overloading the CIMS, while for the TDTs flow rates between 0.02 and 0.2 L min$^{-1}$ are recommended with a total sampling volume up to 4 L. These limits can be included as sampling conditions to stop sampling automatically when the limits are reached. Afterwards the sampling media need to be changed manually. In our experiment, sampling media were changed after typically 1-1.5 h.

After sampling, TDTs are sealed with brass screw caps with PTFE ferrules and filters were kept between precleaned aluminium foil in separate sealed petri dishes. Both are stored at -18 °C in airtight containers until analysis.

## 2.5 Analysis method

The AERTRACC sampling can be used with various kinds of sampling media and consequently can be used in combination with a broad variety of offline analysis methods. For analysis of the samples for this study the TD-HR-ToF-CIMS method was used with the HR-ToF-CIMS (Aerodyne Research Inc., USA) coupled to the FIGAERO inlet for filters and a custom-built inlet for TDT. Iodide served as the chemical ionization reagent which is selective for polar and oxidized organic compounds (Lee et al., 2014). The CIMS allows identification of individual compounds due to soft ionization as well as high-resolution mass spectra. The high sensitivity enables the analysis of small amounts of analyte, minimizing the necessary sample collection times (Aljawhary et al., 2013; Yatavelli et al., 2012).

For ionization, methyl-iodide from custom-made permeation tubes (permeation rate 450 ng/min at 30 °C) is diluted into dry nitrogen (purity 99.9999%), subsequently ionized by an alpha-polonium source (NRD Static Control, USA) to form iodide as reagent ion and inserted into the ion-molecular reaction chamber (IMR) at a flow rate of 2.2 L min$^{-1}$.The filters were thermally desorbed into the IMR with heated dry nitrogen as carrier gas (purity 99.9999%, 1.9 L min$^{-1}$) using the FIGAERO-inlet (Lopez-Hilfiker et al., 2014); TDTs were desorbed with a flow rate of 0,120 L min$^{-1}$ using a custom-built desorption unit. The temperature program for the carrier gas starts at 25 °C for 3 min, heating up to 200 °C with a rate of 17.5 °C min$^{-1}$ and finally holding the temperature for 20 min. Tuning of the ion optics was performed before the first analysis with formic acid and triiodide for signal intensity, mass resolution, and peak shape using the software Thuner (Tofwerk AG, Switzerland). The IMR conditions were kept constant at 130 mbar and 60 °C.

The reproducibility of the integrated ion signal intensity of different calibration compounds, determined through laboratory experiments, was found to be 10% for filter and 62% for TDT samples (details see Sect. S4). Oligomerization during analysis with CIMS might occur (Lopez-Hilfiker et al., 2015) but appeared minor in our testing.

## 3 Characterization of the sampling system

### 3.1 Particle transport efficiency

The aerosol transport losses within the AERTRACC inlet and transport system were estimated with calculations using the Particle Loss Calculator (von der Weiden et al., 2009). The size-dependent transport losses were calculated based on the

geometry of the tubing system considering bends and non-vertical flows as well as volumetric flow rates (Fig. S5). Estimated losses are below 10% for particles between 10 nm and 7 µm in diameter. For particles in the size-range 35 nm up to 3.5 µm, where most of the collected particle mass is typically found, losses are below 2%. Applying the size-dependent losses to a typical urban particle number size distribution, the overall calculated mass losses are below 1 %, both for $PM_1$ and $PM_{10}$. Therefore, we conclude that particle transport losses within the sampling system are generally negligible for the mass-based analysis methods and no correction for losses is needed.

## 3.2 Time delay between aerosol measurement and sampling

In *automatic operation mode*, the AERTRACC sampler is controlled based on the comparison of the specified sampling conditions with the online-measured MoLa data. The difference of the volumetric flow rates between the online instrument and the AERTRACC sampling inlets, which both have the same length and cross section, leads to different aerosol transport times to the instruments and the sampling media, respectively. Due to the higher flow rate through the online instrument inlet of 80 L min$^{-1}$, compared to 30 L min$^{-1}$ (in $PM_1$/$PM_{10}$ *sampling mode*) or 16.7 L min$^{-1}$ (in $PM_1$-only *sampling mode*) through the AERTRACC inlet, the ambient aerosol reaches the online instruments before it reaches the sampling media. This provides the opportunity of knowing in advance whether the aerosol reaching the sampling media should be sampled or not and to switch the sampling valves accordingly.

It is necessary to know the time delay between the online measurement of the aerosol and the aerosol reaching the sampling media to assure timely sampling of the targeted aerosols. The time delay for each instrument is the time difference between the times it takes for the aerosol from the moment it enters the inlet heads until the reporting by the online measurements, and the aerosol reaching the sampling media, respectively.

Self-generated short spikes of elevated aerosol or trace gas concentrations were used to determine the time intervals between the aerosol entering the inlet and the same aerosol being reported by each online instrument for different inlet heights (i.e. 3 m, 6 m, 9 m). These measurements showed that these time intervals can be separated into a transport-related residence time in the inlet tubing and an instrument-specific measurement and reporting delay. The transport-related residence time was extracted from the measurements with different inlet heights, since the instrument-specific measurement and reporting delay is a constant for each instrument and independent of the inlet height. These measured transport times agree well with the calculated transport times of the aerosol, based on tube cross sections and volumetric flow rates. This allows calculating the respective transport times also for the sampling through the AERTRACC inlet without directly measuring it.

In the $PM_1$/$PM_{10}$ *sampling mode* (i.e. with high sampling flow rate) in combination with short inlets of 3 m to 5 m above ground level, for most instruments no delay time must be applied. For instruments with long measurement and reporting time, also no delay needs to be applied even for larger inlet heights.

The time delays for all instruments are implemented in the ACS software for the different inlet heights, which were specified in the *flowrate* sub-window (Fig. 2b). For measurement variables, which are not associated with aerosol transport, like meteorological data or GPS position, the respective instrument time delays are equal to the aerosol transport time through the

AERTRACC inlet. As example, the time delays for the 6 m inlet are 5-17 s for $PM_1$ *sampling mode* and 4-9 s for $PM_1/PM_{10}$ *sampling mode,* excluding instruments with no time delay needed.

For comparison, sampling periods during the in-field validation (see Section 4) were in the order of 2-10 s. Especially under such conditions, where the sampling periods are in the same order of magnitude as the time delays, it is crucial to consider the time delays for sampling. Otherwise, a significant fraction of the aerosol which does not fulfil the various sampling criteria would nevertheless be sampled and the separation of different aerosol types would not be given anymore.

## 4 In-field validation of the AERTRACC using a single point source in a semi-urban environment

### 4.1 Measurement setup

The AERTRACC sampler was tested and validated in the field by probing emissions from a wood-fired pizza oven, operated in a semi-urban environment. The goal was to sample the biomass burning emissions separately from the semi-urban background aerosol using the wind direction and further MoLa variables as sampling conditions. The test setup was located on the premises of the institute (Mainz, Germany), which is located at the outer edge of the city center, on the 21th July 2021. A site map with the measurement location with respect to the city and to the micro-environment including a wind rose plot showing the predominant wind direction can be found in the supplementary information (Fig. S1). The oven was heated with logs of European beech and had a small chimney up to 4 m height above ground level. Larger roads were at a distance of 100 to 150 m, separated by a narrow row of trees and bushes from the measurement site. The main wind direction was northeast to east-northeast with one of the major roads and a fraction of the city upstream of the measurement site. MoLa with the installed AERTRACC sampler was located 13 m away from the pizza oven, in a direction that was frequently downwind of the source. Measurement and sampling inlets were at 4 m height above ground level. Wind was very unstable during the measurement with air arriving temporarily from all directions at the measurement location. Regarding other meteorological parameters, it was a sunny day with few clouds; over the course of the measurement, the temperature was slightly rising from 21 °C to 24 °C while relative humidity decreased from 42% to 35%.

The pizza oven was heated up to 400 °C before pizza baking started. The whole measurement lasted for 3.5 h including 30 min of preparatory measurements to define sampling conditions for separate collection of source emissions and background aerosol.

All MoLa instruments listed in Table 1, including the HR-ToF-AMS with 15 s time resolution, in V-mode for maximum sensitivity (DeCarlo et al., 2006), were operated during the measurements. The flow rates for filter and TDT sampling were set to 5 L min$^{-1}$ and 0.12 L min$^{-1}$, respectively. Filter mass loading was limited to 2 µg and sampling time to 25 min to avoid overloading the filters and exceeding the breakthrough volume of the TDTs. As sampling conditions for the pizza oven emissions, the wind sector 45-90° AND OPC particle number concentrations (PNC) >250 # cm$^{-3}$ were chosen, while for background measurements the conditions were the wind sector 135-360° AND OPC PNC <200 # cm$^{-3}$. Two $PM_{10}$ and two $PM_1$ filters and four TDTs were sampled with pizza oven emissions, and two filters, one for $PM_{10}$ and $PM_1$ respectively, and

two TDTs were sampled with background aerosol. For sampling media blank correction, two filters and TDTs each without sampling were taken as field blanks.

## 4.2 Data Preparation and Analysis

The online data was quality checked, corrected for sampling delays and inspected for invalid data, e.g. data affected by internal calibration procedures, on a 1 s time base. Data with highest available time resolution were used for further data analysis to be able to account for fast wind changes. $PM_1$ mass concentrations were calculated from combined FMPS and OPC size distribution data (details see Sect. S1). The high-resolution AMS data were analyzed with the software SQUIRREL 1.63I and PIKA 1.23I. Furthermore, positive matrix factorization (PMF) (Paatero und Tapper, 1994) was applied on the organic particle fraction below $m/z$ 116, measured with the AMS, using the PMF Evaluation Tool (PET) v3.07C (Ulbrich et al., 2009) to identify different aerosol types. Further details about AMS data processing and PMF are provided in the supplement Sect. S2. For analysis of the CIMS data, the software Tofware 3.2.3 (Aerodyne Inc., USA) and custom data procedures were used (details see Sect. S3). Signal intensity was normalized to the iodide-signal and sampled volumes. Afterwards, the ions signal intensities were averaged over all available samples with pizza oven emissions and background, respectively, both for TDT and filter samples. Data for $PM_1$ and $PM_{10}$ filter samples were handled and analyzed separately. The molecular formula of identified ions was determined for individual peaks; and individual species were identified through the molecular formula, detectability by Iodide-CIMS and previous mention in the literature (further details see Table S1). Signal intensities for individual compounds were determined semi-quantitatively in terms of detected ions as a calibration for each compound was not feasible. This allows determination of relative concentrations in separate samples as well as supporting PMF analysis for quantitative determination of aerosol type concentrations (similar to the approach by Tong et al., 2022). Independently of the sampling media, the ion signal intensities during desorption of the samples exceeded the limit of detection (three times the standard deviation of the molecular background) for all reported samples and ions, with the majority of samples and ions showing an excess by at least an order of magnitude.

## 4.3 Results and discussion

### 4.3.1 Online measurements – characteristics of the measured aerosol

During the field measurement period the AMS provided quantitative data on chemical composition of the non-refractory sub-micron particle fraction. For in-depth analysis of the organic fraction, a PMF analysis was performed for source apportionment. The identified aerosol types were biomass burning organic aerosol (BBOA), cooking organic aerosol (COA) and oxygenated organic aerosol (OOA). This was the most reasonable PMF solution based on the individual PMF factor mass spectra and time series (see Fig. S2). Correlation of the obtained mass spectra with reference mass spectra resulted in average Pearson's r values of 0,86 for BBOA, 0,90 for COA and 0,92 for OOA (Fig. S3). The BBOA mass spectrum shows the typical peaks at $m/z$ 60 and 73, related to levoglucosan as typical biomass burning marker (Schneider et al., 2006). The OOA mass spectrum shows a

380 strong peak for the key marker $m/z$ 44 ($CO_2^+$) from thermal decarboxylation without any further distinct peaks at higher $m/z$ (Ng et al., 2010). For COA no distinct markers exist, except for a high $m/z$ 55 signal (Sun et al., 2011) and the identification was based on comparison with reference mass spectra from the HR-AMS Spectral Database (Ulbrich et al., 2022). The time series of BBOA and COA frequently showed similar temporal variations indicating that they originate from the same source location while the OOA factor was mostly constant over the whole measurement interval and is representing the background

aerosol. Further important time series, like $PM_1$ mass concentration and OPC particle number concentration, are shown in Fig. S6. Time intervals for sampling of source emissions and background are highlighted. Depending on the evaluation of the data, the sampling was frequently (often after only a few seconds or at most minutes) switched between source and background aerosol paths.

Because of the short measurement time and the close vicinity to the source, the temporal variations of aerosol and trace gas

concentrations were mainly due to changes in wind directions and variations in emission strength of the targeted source rather than to those of other sources or of atmospheric dilution. In Fig. 3a the concentrations of the three organic aerosol types, i.e. PMF factors, are shown as a function of the wind direction, averaged over 15° wind sectors. Further aerosol concentrations, which are assumed to be associated with the background and source emissions, are shown in Fig. 3b with suitable scaling factors to plot them together in a single polar graph. A strong dependence of mass concentration for BBOA and COA on wind

direction with a maximum for wind from the sector 60° to 90° was observed (Fig. 3a). A similar dependence on wind direction was found for black carbon (BC) and polyaromatic hydrocarbons (PAH) (Fig. 3b), which are also likely associated with emissions from the pizza oven as well as BBOA and COA (Fachinger et al., 2017). OOA, as an indicator of background aerosol, is almost constant for all wind directions as well as sulfate ($SO_4$) which is often an indicator for secondary oxidized aerosol (Sun et al., 2011). These results show a clear enhancement of concentrations of aerosol components, which are related

to the pizza oven emissions, when the wind was arriving from the direction of the source, which was located in the direction of 70° with respect to the sampling location.

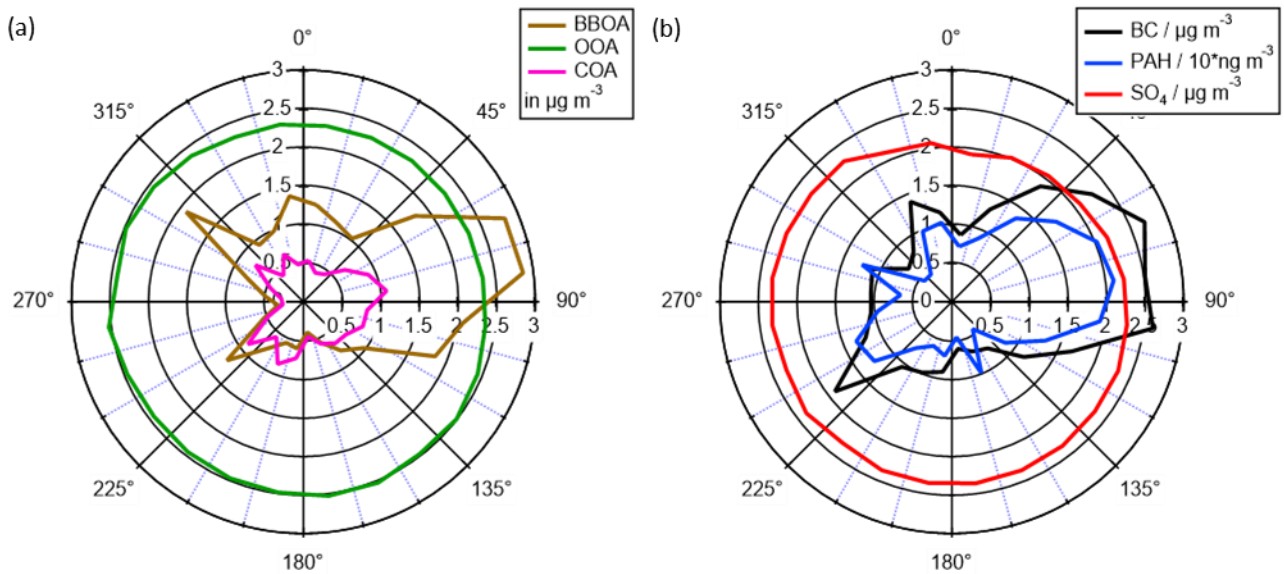

**Figure 3: Concentrations of the organic aerosol types (a) as well as BC, PAH and SO₄ (b) dependent on local wind direction averaged over 15° sectors. The Pizza oven was located in the direction of 70° relative to MoLa.**

### 4.3.2 Filter and TDT analysis

Source and background aerosol were separately sampled on filters and TDTs with sampling conditions based on preparatory measurements (see Sect. 4.1). The comparison of averaged signal intensities for identified ions from $PM_1$ and $PM_{10}$ filter samples showed only negligible differences (Fig. S7), suggesting that most of the related aerosol mass is in the $PM_1$ particle size range. Therefore, the results are discussed for the $PM_1$ filters only.

The ratio of the ion signal intensity for selected identified species from the pizza oven and the background samples was calculated for the filter and the TDT samples (Fig. 4), respectively, to show which of the species mainly originate from background and which ones are associated with the source emissions. Additionally, the average ratio for all species assigned to only background (*aged/traffic*) and oven emissions (*biomass burning/cooking – BB/C*) as well as both groups (*mixed*) were calculated for comparison. The assignment to the sources must be regarded as a rather preliminary one, as the apportionment is only based on a literature search. The list of identified species and used acronyms is shown in Table 2 and Table 3 for the filter and TDT samples, respectively. Substances found on the filters and TDTs differ mainly due to gas-particle partitioning and the selectivity of the TDT adsorbents. Volatilization of material from the filters and subsequent sampling in the TDTs could lead to biased information on the partitioning of substances, however, within the uncertainties of the analysis, this effect is presumably not significant.

For some ions, based on the molecular formula, several substances are possible which are listed as well. Details like the exact *m/z* of the ions and references for source apportionment of the species are summarized in Table S1 and S2. The ratio is expected to be on the order of one for species, which originate from background aerosol only. They are typically associated with aged,

oxidized aerosol or traffic emissions and should be found on the background and source samples in roughly equal amounts, after correcting for sampled volumes, since their origins are well distributed over all wind directions (see also Fig. 3, OOA aerosol). This is the case for the species found on the filter samples (Fig. 4a) that were assigned to traffic emissions or aged aerosol.

In contrast, identified compounds from the filter samples with source-to-background intensity ratios significantly larger than one are mostly known to be associated with biomass burning and cooking emissions, which is in good agreement with their higher abundance on the pizza oven-related filters. Compounds like levoglucosan (LG) and pyroglutaminic acid (PGA) which are markers for biomass burning and cooking, respectively, show more than 85 times higher intensities on the source-related filters compared to the background filters.

Based on a literature search, some of those species, associated with cooking and biomass burning, can also originate from various other emission sources and were therefore assigned to the *mixed* group. They have a variety of different ratios between 0.6 and 10, showing that probably some of them predominantly originate from the background aerosol while others mainly from the pizza oven emissions.

The large average source-to-background ratio for compounds attributed to biomass burning and cooking shows that the targeted source emissions from the pizza oven were sampled predominantly on the source-related filters and not or only to a small degree on the background filters. Compared to that the average ratios for the aged and traffic related compounds as well as the *mixed* aerosol are considerably smaller indicating a clear separation of source-related emissions from background-only aerosol using the selected AERTRACC sampling criteria.

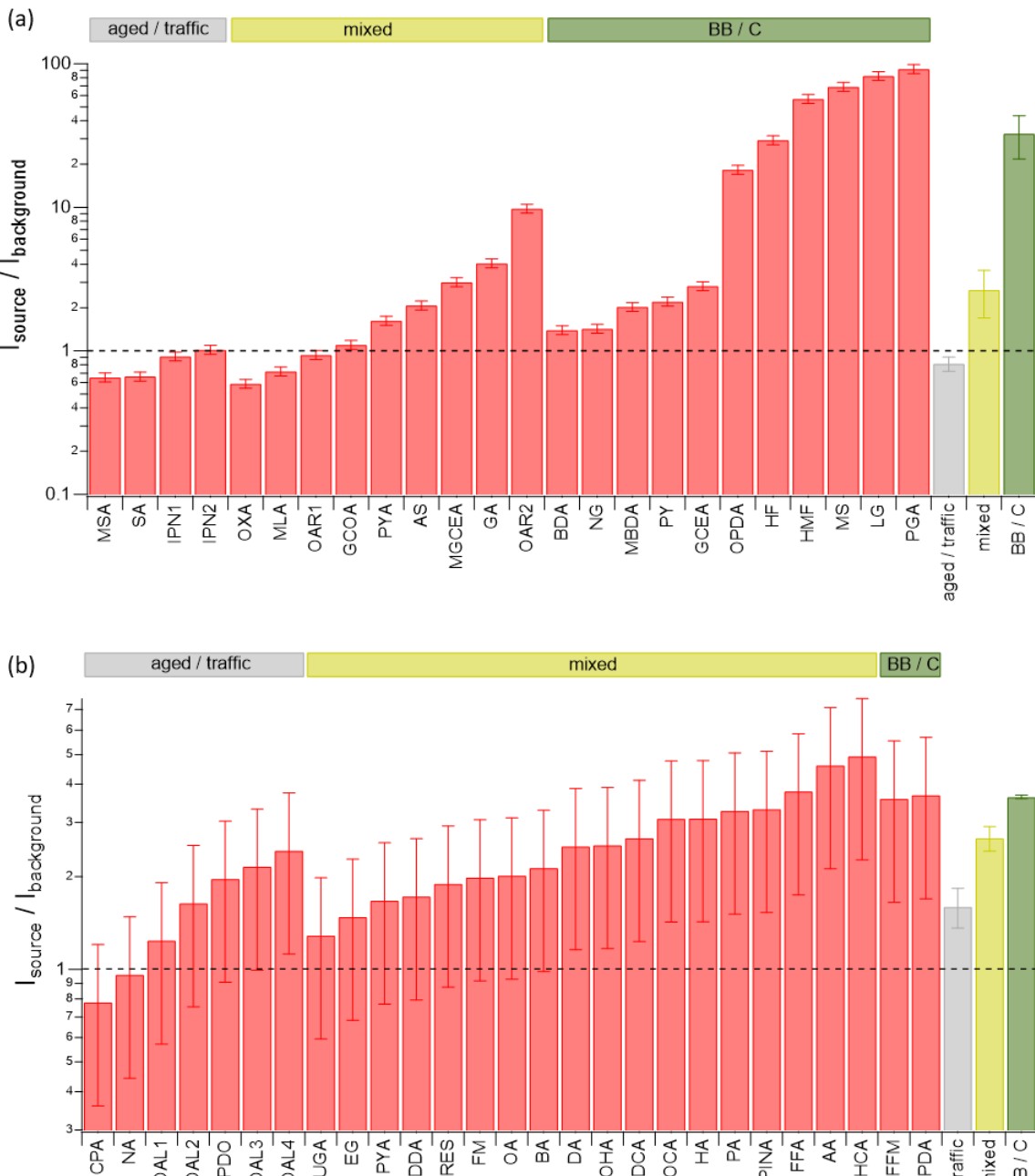

**Figure 4: Ion signal intensity ratios of identified compounds from pizza oven and background for filter (a) and TDT (b) samples with source apportionment based on references (see Table S1 and S2). The dashed line represents a ratio of one, i.e. similar intensities found on pizza oven and background samples. The abbreviations *BB* and *C* stand for biomass burning and cooking. The errors bars are based on the standard deviation of the ion signal intensity, reproducibility, and the error obtained from the blank measurements (for details see Sect. S4).**

**Table 2: Selected identified compounds, measured as iodide cluster, from filter analysis and acronyms used for Fig. 4a. For further details see Table S1.**

| Acronym | Assigned compound | Acronym | Assigned compound |
|---------|-------------------|---------|-------------------|
| AS | ascorbic acid, hydroxyfurans | MLA | malic acid |
| BDA | butenedioic acid | MS | monosaccharide |
| GA | glutaric acid | MSA | methanesulfonic acid |
| GCEA | glyceric acid | NG | nitroguaiacol |
| GCOA | glycolic acid | OAR1 | oxidized aromats, 3-acetylpentanedioic acid |
| HF/FA | hydroxy furfural, furoic acid | OAR2 | oxidized aromats |
| HMF | hydroxymethyl furfural | ODPA | 2-oxopropanedial, oxoacrylic acid |
| IPN1 | oxidized isoprene nitrate | OXA | oxalic acid |
| IPN2 | oxidized isoprene nitrate | PGA | pyroglutamic acid |
| LG | levoglucosan, galactosan, mannosan | PY | pyranose |
| MBDA | methylbutendioic acid | PYA | pyruvic acid |
| MGCEA | methylglyceric acid | SA | sulfuric acid |

**Table 3: Selected identified compounds, measured as iodide cluster, from TDT analysis and acronyms used for Fig. 4b. For further details see Table S2.**

| Acronym | Assigned compound | Acronym | Assigned compound |
|---------|-------------------|---------|-------------------|
| AA | acetic acid | OA | octanoic acid |
| BA | butyric acid, methyl propanoate | OAL1 | oxidized alkyl |
| CHCA | cyclohexenecarboxylic acid | OAL2 | oxidized alkyl |
| CPA | β-caryophyllene-aldehyde | OAL3 | alkyldiole |
| CRES | cresol | OAL4 | oxidized alkyl |
| DA | decanoic acid | OCA | oxocarboxylic acid |
| DCA | decenoic acid, pinanediol, linalool oxide | ODPA | oxopropanedial, oxoacrylic acid |
| DDA | dodecanoic acid, methylundecanoic acid | OHA | oxohexanoic acid, ethyl acetoacetate, methyloxopentanoic acid |
| EG | ethylene glycol | PA | propanoic acid |
| FFA | furfuryl alcohol, 2-furanmethanol | PDO | propandiol, hydroxyacetone |
| FFM | N-formylformamide, nitroethen | PINA | pinalic-3-acid |
| FM | formamide | PYA | pyruvic acid |

| HA | hexanoic acid, cyclopentanoic acid | SUGA | sugar acid |
|----|-----------------------------------|------|------------|
| NA | nonenoic acid | | |

Only two identified compounds from the TDT analysis were attributed solely to source-related emissions, i.e. cooking and biomass burning, and both substances have ratios well above one as they probably originate from the pizza oven emissions (Fig. 4b). The compounds assigned to traffic and aged aerosol have partially ratios on the order of one but also partially significantly above one, i.e. they are present on source-related TDTs in larger amounts than on background-related TDTs. Either these compounds are emitted by a close unknown source located in the same wind direction as the pizza oven or they are emitted by the pizza oven as well and thus would belong to the mixed group. Most of the identified compounds from the TDT samples can be assigned to different sources (*mixed*) having ratios which can be related to background aerosol and also to source related emissions.

Compared to the filter analysis the difference between average ratios of all source- and background-related compounds from the TDT analysis is smaller suggesting a weaker separation of source and background emissions. However, it must be taken into account that few compounds were assigned to only one of the aerosol types. As most of the compounds can originate from background as well as source-related emissions the enrichment of source-related compounds is smaller if these compounds are already present in the background aerosol. Thus, no specific markers were identified for the gas phase of the pizza oven emissions, which would clearly show a very strong difference between background and source-related TDTs, in contrast to e.g. levoglucosan and pyroglutaminic acid on the filter samples.

In conclusion, for the filter samples the chosen sampling conditions for the background and source emissions proved to be suitable to sample the source emissions separately while the background emissions are found in approximately equal concentrations on the source and background filters at least based on the identified compounds. A gravimetric analysis of the samples could be performed in addition to the chemical analysis to extend the general information on the sampled aerosols. For the TDT samples the shown ratios indicate a weaker separation of source and background emissions, likely because most of the identified compounds can originate from both, background and source emissions, and no distinct markers were found for the source emissions.

### 4.3.3 Evaluation of sampling conditions

The highly time-resolved MoLa online data provide the opportunity to post-evaluate the chosen AERTRACC sampling conditions. This is done by comparing average source-related and background aerosol concentrations as well as total source-related sampling time for the chosen and other potential sampling conditions and by evaluating, whether a better separation between source emissions and background could herewith have been achieved. The selected separation for the pizza oven measurement was based on a combination of PNC measured by OPC and wind direction (*Wind+OPC*), see Table 4 for details. For comparison, simpler conditions using only the wind direction (*Wind*) and stable wind conditions (*Wind stable*) were evaluated. Stable wind conditions are fulfilled when wind from the source sector was observed at least for the previous 8 s, the

transport time from the source to the MoLa inlet, which was calculated from the distance between the measurement inlet and the pizza oven, and the average wind speed during the measurements. The combination of PNC measured by CPC and wind

direction was evaluated as additional sampling scenario (*Wind+CPC*). Further sampling conditions were defined based on the AMS data using fractions of the organic signals at single *m/z*, e.g. at *m/z* 55 as $f_{55}$, to test whether a potential use of the AMS for AERTRACC control could improve aerosol separation. The selection of a combination of wind direction and $f_{55}$ (*Wind + $f_{55}$*) as well as $f_{55}$ and the ratio $f_{55}/f_{57}$ (*Wind + $f_{55}$ + $f_{55}/f_{57}$*) was based on known markers for COA while the combination of wind direction and $f_{60}$ (*Wind + $f_{60}$*) was based on the known marker for BBOA. The limit values in the sampling condition

definitions were chosen from literature values for these aerosol types (Cubison et al., 2011; Elser et al., 2016; Mohr et al., 2009; Mohr et al., 2012; Saarikoski et al., 2012; Sun et al., 2011; Xu et al., 2020).

The mass concentrations of black carbon (BC), polyaromatic hydrocarbons (PAH), organics measured by AMS, the AMS PMF factors BBOA, COA, and OOA, $PM_1$ as well as PNC measured by CPC and OPC were used to compare how well different sampling scenarios separate between source emissions and background. These parameters were chosen as they

showed to be strongly affected by the source emissions during the measurement, according to the online data analysis (Sect. 4.3.1).

**Table 4: Sampling conditions for compared sampling scenarios for source and background sampling.**

| Sampling scenario | Source | Background |
|---|---|---|
| Wind | Wind direction 45-90° | Wind direction 135-360° |
| Wind stable | Wind direction 45-90° for 8 s | Wind direction 135-360° for 8 s |
| Wind + CPC | Wind direction 45-90° AND CPC PNC > 20,000 # cm$^{-3}$ | Wind direction 135-360° AND CPC PNC < 15,000 # cm$^{-3}$ |
| Wind + OPC | Wind direction 45-90° AND OPC PNC > 250 # cm$^{-3}$ | Wind direction 135-360° AND OPC PNC < 200 # cm$^{-3}$ |
| Wind + $f_{55}$ | Wind direction 45-90° AND $f_{55}$ > 0.07 | Wind direction 135-360° AND $f_{55}$ < 0.05 |
| Wind + $f_{55}$ + $f_{55}/f_{57}$ | Wind direction 45-90° AND $f_{55}$ > 0.07 AND $f_{55}/f_{57}$ > 2 | Wind direction 135-360° AND $f_{55}$ < 0.05 AND $f_{55}/f_{57}$ < 1.5 |
| Wind + $f_{60}$ | Wind direction 45-90° AND $f_{60}$ > 0.01 | Wind direction 135-360° AND $f_{60}$ < 0.005 |

For assessment of source and background aerosol separation based on the various sampling scenarios, the ratios of averaged concentrations for "source" and "background" intervals, i.e. when the respective conditions were fulfilled, were calculated for each variable and each scenario (Fig. 5a). In addition, the potential sampling times that would have been spent to sample the source emissions and background aerosol for the various sampling scenarios, are shown in Fig. 5b.

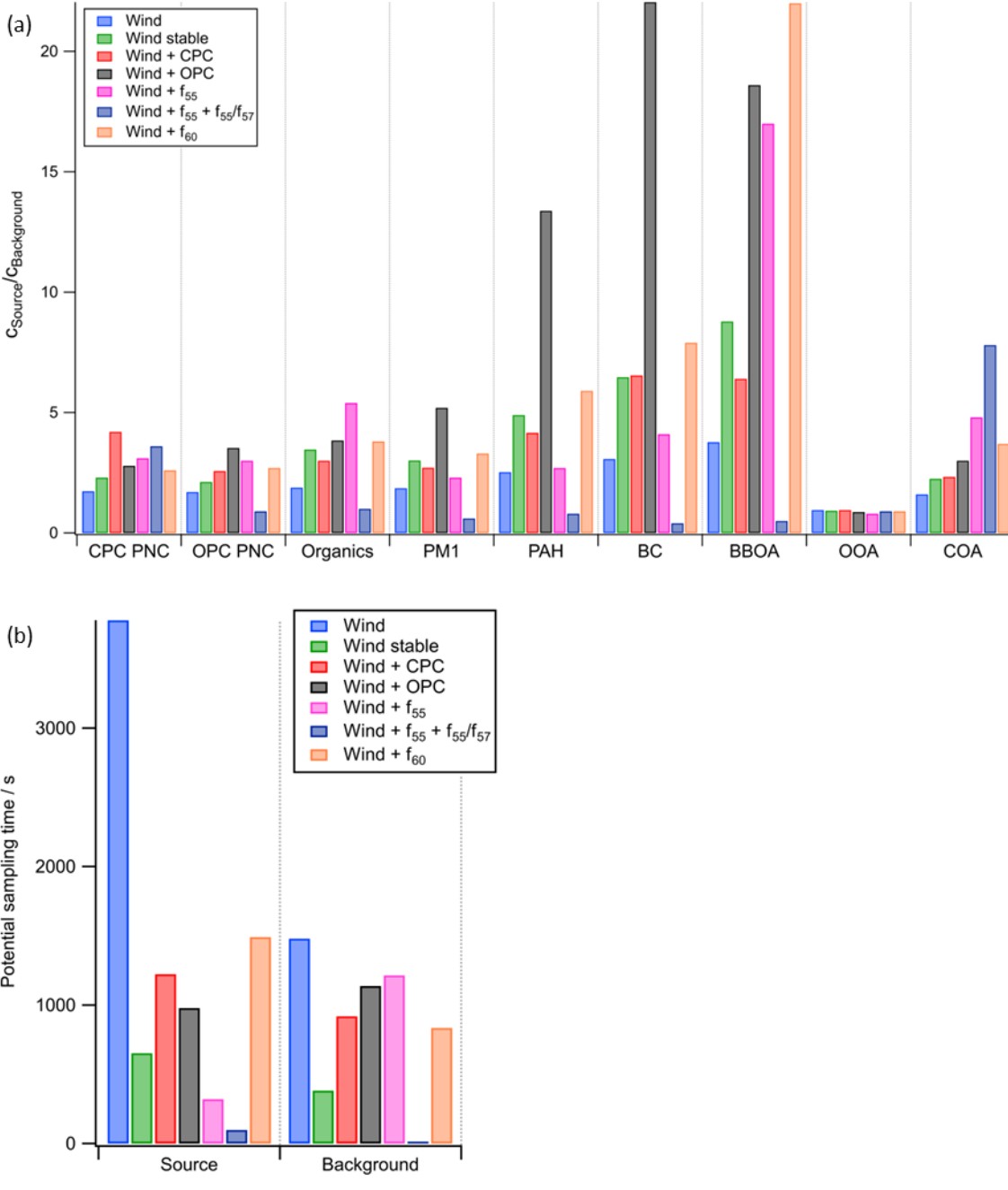

Figure 5: Ratio of averaged mass concentration and PNC of "source" and "background" aerosols, according the seven different sampling scenarios (a), and related potential source emission and background aerosol sampling times (b).

Using only wind direction as separation criterion leads to the longest sampling times, especially for the source-related sampling. However, this approach also results in the smallest ratios of source versus background concentrations, i.e. the least

effective separation of source emissions and background. Both effects are the result of the unspecific definition of the sampling condition.It is possible that source emissions miss the sampling inlet due to fast wind changes, which then samples background or mixed aerosols, even though the "source" sampling criterion is fulfilled. Using stable wind conditions as sampling scenario improves the separation substantially, but at the expense of sampling time, which is by far the lowest for all four sampling scenarios.

The combination of elevated CPC PNC and the wind direction as sampling condition leads to higher ratios for measured CPC PNC and $PM_1$ compared to the *Wind stable* sampling scenario, but similar or smaller ratios for the other parameters. The sampling time is longer than for *Wind stable*, however still much lower than for the *Wind* sampling condition.

The largest ratios for almost all variables besides the AMS-based ones, and consequently the most effective separation of source-related and background aerosol, were achieved when elevated PNC measured by OPC additional to the right wind

direction were used as sampling condition (*Wind+OPC*). This sampling scenario resulted in similar sampling time as the other "complex" sampling scenario *Wind+CPC* and strongly improved sampling time, compared to the *Wind stable* scenario. Improved measurement of particle mass-related variables like $PM_1$ or PAH mass concentration in this sampling scenario occurs, since the OPC counts the larger particles ($d_p = 0.25\ \mu m - 32\ \mu m$) and therefore the OPC PNC represents the emitted mass concentration quite well. The CPC, on the other hand, counts smaller particles ($d_p = 5\ nm - 3\ \mu m$); therefore, it captures

better the total emitted PNC with the very small particles contributing little to the emitted mass. Since for analysis of the sampling media, sampled particle mass is the more relevant variable, compared to particle number, the *Wind+OPC* sampling scenario is better suited to control the AERTRACC, compared to the *Wind+CPC* scenario. Contrary, in case of new particle formation events, the freshly formed aerosol could be targeted using high CPC PNC and low $PM_1$ concentrations or low OPC PNC as sampling conditions.

Inclusion of the AMS data in the AERTRACC control using the fractional signal intensity of known marker m/z could improve specific sampling for certain aerosol types. This is especially the case if the AMS is operated with shorter averaging intervals to capture short-time variations of air masses containing different aerosol types. For COA, higher source/background ratios were achieved with the *Wind + $f_{55}$* sampling scenario, compared to the other scenarios, and even higher ones with the *Wind + $f_{55}$ + $f_{55}/f_{57}$* scenario as it is more specific for COA. Regarding the potential sampling times especially within the latter scenario,

the times are quite limited due to the very specific conditions and possibly due to shorter COA emission periods compared to the more dominant BBOA. The *Wind + $f_{60}$* scenario enables the most effective separation for BBOA combined with potential sampling times comparable to the Wind+OPC scenario.Long potential sampling times are desirable in order to quickly collect the necessary mass or sampling volume for analysis. Therefore, for scenarios like *Wind stable* and *Wind + $f_{55}$ + $f_{55}/f_{57}$*, longer overall measurement periods in the vicinity of the source are necessary to reach sufficient sampled aerosol mass.

The choice of smaller wind sectors within the originally chosen wind sector 45-90° was evaluated in an additional analysis to investigate whether this could improve (i.e. enhance) the ratio between average source and background concentrations,

compared to the *Wind+OPC* scenario. The calculated ratios for all variables for the splitting of the original wind sectors into three, five and seven sectors are shown in Table S3-S5. The split into three sectors improves the separation of source and background emissions for the middle sector in comparison to the *Wind* scenario by at maximum 13 %. Further splitting leads to partially improved ratios between source and background emissions by at maximum 20 % for five sectors and by at maximum 22 % for seven sectors. However, the maximum values of ratios for different measured parameters spread over several wind sectors and therefore does not point towards a "better" potential selection of the source wind sector. This spread is probably due to indirect transport of the aerosol to the inlet due to frequently changing wind directions as well as due to different time resolutions of the instruments. Additionally, with decreasing width of the wind sectors, the potential sampling time per sector decreases for all sections leading to longer overall measurement times necessary to sample sufficient amounts for subsequent analysis. Despite the improvement through smaller wind sectors, the ratios of the *Wind+OPC* scenario were by far not reached, and the source-related sampling times were shorter for the 5- and 7-sector splitting, compared to the *Wind+OPC* scenario. Consequently, using narrower wind sectors does not improve the separation of source and background emissions as effectively and as efficiently as choosing additional parameters to define the sampling conditions. In addition, using only narrow wind sectors for separation of source-related and background aerosol requires very good knowledge about the wind direction for which the emission source is probed. This is not the case when wind direction is used in combination with other emission source-related features of the aerosol as sampling criterion. Therefore, in general, source-specific markers are needed, which are known and can be measured by MoLa, to define source-specific sampling conditions and to achieve the separate sampling of these emissions.

**5 Summary**

We developed the sampling system AERTRACC (AERosol and TRACe gas Collector) to separately sample the particulate and gas phase of source emissions and background aerosol in complex environments. It is incorporated in our mobile laboratory (MoLa) with its own inlet. Up to four samples can be taken in parallel; in this study, each sample was taken onto a filter and a thermal desorption tube (TDT) for the particle and gas phase, respectively. Separation of different aerosol types is achieved through external control of the sampler based on online measurements of MoLa by setting suitable sampling conditions for the individual aerosol types, which are compared with the online data. An in-house developed software is implemented in the MoLa data acquisition software for direct data access. For each of the four sampling paths up to four measured variables can be combined to create sampling conditions for the targeted aerosol type, which are continuously compared with the current measured data. Besides the automatic sampling, the sampler can also be controlled manually.

The inlet and transport system was designed for minimal particle losses with typical estimated mass losses below 1 % for particles in the size-range 35 nm up to 3.5 µm. Due to shorter residence time of the aerosol in the MoLa online measurement inlet, compared to the sampling inlet, it can be analyzed with the online instruments and the sampling conditions are evaluated

before the aerosol reaches the sampling media. These time delays were experimentally determined for all instruments and are considered in the AERTRACC control software.

For proof of concept and in-field validation, pizza oven emissions were probed in a semi-urban environment. The CIMS analysis of the hereby collected filters showed the successful separate sampling of source emissions from the background aerosol. Compounds known to be related to biomass burning and cooking were predominantly found on the source emissions filters while compounds associated with aged aerosol or traffic emissions were found in similar amounts on the background filters and the source emission filters. For gaseous species, the analysis of the TDTs indicate only a weak separation of source

and background emissions mainly because most of the identified species can originate from aged and traffic aerosol as well as from biomass burning and cooking emissions and no distinct markers were identified for the pizza oven emissions. Hence, these compounds can already be present in the background aerosol leading to a smaller increase in their concentrations due to source emissions.

The comparison of different potential sampling scenarios demonstrated the advantage of combining different measured

variables to achieve targeted sampling of desired emissions. The separation using solely wind direction as sampling criterion was weak due to varying wind conditions leading to nonlinear aerosol transport. Adding source specific criteria like elevated particle number concentrations measured by the OPC improved the separation. As a consequence of this more effective separation of the emissions, the source apportionment of identified compounds is improved. A future addition of AMS for AERTRACC control would offer the possibility to define specific sampling conditions for certain aerosol types, like BBOA

and COA, derived from AMS measurements, using known markers.

An important requirement for AERTRACC to sample targeted aerosol types is the knowledge about the source aerosol properties, which can be determined in preparatory measurements to define suitable sampling conditions for the different aerosol types. Under such conditions, AERTRACC is capable to separate emissions of individual sources from those of other sources or from the aerosol background for improved chemical analysis of source-related emissions even in complex

environments. Possible complex situations could be an industrial facility, like a steel plant, with different but closely located emission sources, e.g. coke oven, blast furnace, sinter plant, and traffic; or urban environments with emissions from traffic, wood combustion, and restaurants. Apart from TD-CIMS, a broad variety of chemical, physical, and microscopicalanalysis methods could be used in combination with AERTRACC to acquire the desired kinds of information from the samples.


*Author contribution.* JP and FD conceptualized the sampling system and field measurement. JP carried out the experiment, analyzed the MoLa data and prepared the paper with contributions from FD, LM und SB. LM developed the CIMS method and analyzed the samples using the developed method.

*Competing interests.* The authors declare that they have no conflict of interest.

*Acknowledgements.* We thank Thomas Böttger, Philipp Schuhmann, Antonis Dragoneas and the mechanical workshop for great support in the technical realization of the sampler. We also thank David Troglauer and Carsten Pallien for support during the in-field validation. Furthermore, we acknowledge the Max Planck Institute for Chemistry for funding of this work.

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
