# Peer review of "The AERosol and TRACe gas Collector (AERTRACC): an online measurement controlled sampler for source-resolved emission analysis"

_Atmospheric Measurement Techniques, 2022_

## Author Comment (AC1)

Dear reviewer,

we thank you for your valuable feedback to our manuscript, which we believe makes it more valuable to the readers. Within the letter, we commented on and replied to all suggestions and requests. The changes made to the manuscript are written italicized in quotation marks within the reply. Within the manuscript, we marked all changes with track changes.

**Reply on "Comment on amt-2022-206" of Anonymous Referee #1**

The submitted manuscript by Pikmann et al introduces a new, automated offline sampler for both aerosols and gases called the AERTRACC, which is part of the mobile laboratory platform MoLA. The stated aim in its design is to allow for more efficient source separation when sampling complex source environments, specifically for the analysis of organic aerosol (OA) and their precursors. The manuscript describes the hardware and control software in detail, and then demonstrates its operation with a (static) single source experiment. That includes an in-depth analysis of gas and particulate phase tracers detected by their analyzer of choice (an TD-CIMS) and the source-specific contrast ratios observed. The authors conclude with a detailed discussion of the optimal choice of input parameters to control the sampler, in order to increase source contrasts for this particular case. So while there is no data shown from actual mobile sampling, the reader is reasonably reassured that it will in fact work for that scenario.

The overwhelming majority of offline samplers operate on some type of predefined timegrid, which leads to different airmasses and hence sources and chemistries being integrated into one sample. For mobiles sources where air mass variability is higher this can limit the usefulness of offline analysis significantly. Hence, on airborne platforms, most offline samplers (such as whole air samplers) are operated based on some type of expert assessment of airmass change, which in (some, rare) cases is automated. While aircraft payloads are highly variable from mission to mission, that is obviously not the case for the MoLA platform, which has a well-defined instrument payload. Therefore, it makes sense to not only build a "smart" offline sampler for MoLA, but to automate it based on all the possible ancilliary measurements available. So while this type of sampling has been done before, this paper shows a technically proficient, well-executed implementation of it for a platform that with certainly make good use of it in the future. So this is well within the scope of and of general interest to the AMT audience. However, I have some concerns about the way the work is framed and about the details of the analysis, which are described in the following:

Unless I am completely misunderstanding the work presented, the key aspect of this sampler is that it switches between "source" and "background" conditions many, many times over the course of the experiment. This is quite different from most offline samplers were only one continuous sample is taken. I would recommend making this much clearer in the description, and possibly illustrating this by e.g. showing the actual sample times on top of Figure S5. It would also be good if the authors discussed what the minimum sample time (per valve switch) actually is, and how this could possibly impact the overall performance once MoLA is actually moving and hence the source sector segments become smaller.

**Answer:**

Yes, this is correct. Thank you for your suggestions to clarify the operation principle. We added the actual sampling intervals for the source and background aerosol to Figure S6 (previously S5) and changed the figure caption accordingly. We also added a sentence about the sampling periods and the switching between the sampling paths based on the evaluation (Section 4.3.1):

"Time intervals for sampling of source emissions and background are highlighted. Depending on the evaluation of the data, the sampling was frequently (often after only a few seconds or at most minutes) switched between source and background aerosol paths."

Further, we added a short description in the section on sampler operation (Section 2.3) in which we describe that during an experiment the sampler typically switches multiple times between different sampling paths, based on which aerosol type conditions are fulfilled, and discuss the impact on mobile sampling experiments.

"During measurements when air masses containing different aerosol types reach the inlet, the sampler switches automatically between the according sampling paths based on the evaluation of the sampling conditions each second. Therefore, switching between different sampling paths typically occurs multiple times within an experiment of hours of duration, which is in contrast to conventional continuous sampling. Although the AERTRACC is primarily designed for stationary measurements, it is also possible to sample during mobile measurements if the air mass segments are large enough to differentiate between them on a few seconds time scale."

Related to the last item, can the authors please discuss what the limitations are on leaving the "source" channel idle for long periods between source interceptions? My specific concern is that any type of filter substrate has issues with loss of volatiles (e.g. Heim et al, 2020), and while during active sampling these "losses" will likely end up in the TDTs, once the line is idle they will simply evaporate and possibly diffuse/deposit to the "background" channel. And there is also, specifically for CIMS measurements, obviously the oligomerization problem (e.g. Lopez-Hilfiker et al, 2015), which should at least be acknowledged.

**Answer:**

We agree that losses of volatiles from filters are always possible and that those volatiles will likely be adsorbed by the TDTs during sampling, if the types of molecules can be adsorbed by the adsorption media. During times when the sampling line is idling, volatiles will diffuse away from the filter. A fraction of those will end up in the TDT. Diffusion and deposit into other sampling lines, however, is not possible, as the aerosol would have to diffuse against the non-sampling path flow (which is active during non-sampling conditions, see Figure 1) up to the next flow split.

To clarify this, we added the following sentence to Section 2.2:

"Further, for non-sampling conditions, diffusion of volatiles from one sampling path to another is avoided as the volatiles would have to diffuse a short distance upstream the flow persisting through the non-sampling path."

We agree that oligomerization might occur during the analysis, however this was not observed for our measurements. We added this information to the manuscript as suggested (Section 2.5).

"Oligomerization during analysis with CIMS might occur (Lopez-Hilfiker et al., 2015) but was not observed within this study."

I do not think that the current introduction does the best job in framing the significance and applicability of AERTRACC. First of all, I think the sampler needs to be separated more clearly from the specific application (in this cases, molecular OA composition). So by way of example, the current AERTRACC, with a simple change in filter media would surely be of high interest to single-particle TEM folks, a field where optimizing contrast for the source-specific aerosol types of interest (and not necessarily organic ones) can be extremely challenging. So framing AERTRAC as an efficient way of solving the contrast issue for any type of off-line analyzers will better illustrate its potential.

**Answer:**

We agree that it would be better to focus more on the AERTRACC sampler and to stress that the application of the sampler is not limited to molecular OA composition analysis but applicable to all kinds of offline analysis methods. Therefore, we changed the last paragraph of the introduction accordingly.

"AERTRACC collects samples of different aerosol types for subsequent in-depth analysis on separate sampling media which can be quickly and simply exchanged."

**and**

"While online instruments for in-depth chemical analysis with high temporal resolution are limited to the respective analysis methods, the AERTRACC sampler enables the use of the full potential of analytical chemistry and microscopic analysis for the investigation of such aerosols beyond these specific approaches. For this work, TD-HR-ToF-CIMS (thermal desorption high resolution time-of-flight chemical-ionization mass spectrometry) was chosen as analysis method offering high resolution mass spectra combined with high sensitivity and low sample fragmentation as well as minimized sample preparation effort (Aljawhary et al., 2013; Mercier et al., 2012; Yatavelli et al., 2012)."

In addition, we added a short sentence in the beginning of Section 2.4 that the choice of sampling media was suitable for the CIMS but is not limited to it.

"Generally, the sampling media used are dependent on the subsequent offline analysis method. The choice of the sampling media for this study was based on the selection of thermal desorption as sample introduction method for the subsequent analysis using TD-CIMS, which reduces the chances of potential contamination through sample preparation."

Furthermore, we added a sentence in in the beginning of Section 2.5 to stress that AERTRACC can also be used for other offline methods.

"The AERTRACC sampling can be used with various kinds of sampling media and consequently can be used in combination with a broad variety of offline analysis methods."

The summary was also adjusted based on the reviewer's suggestion.

"Up to four samples can be taken in parallel; in this study, each sample was taken onto a filter and a thermal desorption tube (TDT) for the particle and gas phase, respectively."

**and**

"Apart from TD-CIMS, a broad variety of chemical, physical, and microscopical analysis methods could be used in combination with AERTRACC to acquire the desired kinds of information from the samples."

More importantly, this statement (L79-80) "Currently, no instrument offers detailed chemical analysis of aerosols in real-time for the analysis of individual sources (Parshintsev and Hyötyläinen, 2015)" which is the linchpin of the current introduction, is simply incorrect. There are real-time, 1 Hz capable molecular OA detectors, such as the EESI-ToF (Lopez-Hilfiker et al, 2019, Pagonis et al, 2021) or the CHARON-PTR-MS (Eichler et al, 2015, Piel et al, 2019). None of them are perfect (no instrument is), but the point is that molecular identification of OA from mobile sources can be accomplished in many different ways, and AERTRACC+TD-CIMS is just one of many, and not really the main reason (in my view) for AERTRACC.

**Answer:**

Thank you for this comment. We agree and removed the mentioned statement. Instead, we added the mentioned instruments as current high time resolution instruments.

"A few instruments with high time resolution in the order of seconds, sufficient for the analysis of transient aerosol occurrences, combined with detailed analysis were developed in recent years, such as the EESI-ToF (electrospray ionization time-of-flight mass spectrometer) (Lopez-Hilfiker et al., 2019; Pagonis et al., 2021) and the CHARON-PTR-MS (chemical analysis of aerosol online proton-transfer-reaction mass spectrometer) (Eichler et al., 2015; Piel et al., 2019)."

What both the introduction and the rest of the paper do not discuss are detection limits, which are typically one of the main design criteria for samplers. This needs to be addressed, since currently there is not context whatsoever on what a reasonable total sample time would be (and hence how practical the use of AERTRACC is for typical urban source applications). Based on the description, under typical polluted urban conditions (~10 ug m-3 OA), it would take about 2 hours to sample 1 ug of material, which given typical DLs for organic compounds in the FIGAREO (~0.5 ng sm3 from filter blanks) seems sensible. But that seems like a long sampling time for a local source, so a discussion of the possible tradeoffs would be appreciated.

**Answer:**

Thank you for this comment, we agree that this needs further discussion. Indeed, for CIMS analysis we aim to sample in the order of 1-2  $\mu$ g of material to avoid overloading the CIMS (as stated in Section 2.4). For other chemical analysis methods, the desired amount of sample material is in a similar order of magnitude, while for microscopic analysis it could be much lower. At an ambient concentration of 10  $\mu$ g m-3 with a sample flow of 7.5 LPM, this translates to a sampling time of 13 min, which needs about two hours of measurements if only 10% of the time (as during our validation experiment) the emissions are sampled. Therefore, to collect sufficient sample material, it must be possible to probe the air in the vicinity of the source over sufficiently long time. This also means that both, the source emissions and the ambient transport conditions must prevail over this time. The chemical analysis of

single events, short or very weak emissions is therefore likely not possible with this setup. The use of microscopic methods for the analysis of the samples, however, might strongly reduce the necessary sampling times or source concentrations, for successful analysis.

We added a discussion on this issue at the end of Section 2.3 on sampler operation:

"The chemical analysis of aerosol samples (like e.g. when using FIGAERO-CIMS measurements of organic compounds) typically requires sampled mass in the order of 1  $\mu$ g in order to exceed instrumental detection limits, depending on the specific analysis method. In polluted urban conditions with organic mass concentrations of 10  $\mu$ g m-3, with a sample flow rate of 7.5 L min-1 and with approximately 10 % of the time sampling source emissions (like in our validation experiment, see Section 4), a total sampling time in the order of two hours would be needed to collect enough material for analysis. Therefore, the probed source must emit over sufficiently long times to allow a successful chemical characterization of their emissions. Higher emission concentrations, more stable transport conditions, and lower detection limits of the applied analysis method can reduce sampling times significantly. Especially when using microscopic and single particle techniques, which might need extremely low amounts of sample, sampling times could be reduced further and also single transient emission events might provide sufficient material for successful analysis."

The (nicely done) PMF analysis of the AMS measurements seems a bit underutilized at the moment. Its sole purpose currently is to show the wind dependence for the OA sources, which could certainly be showcased with less work. Obviously it is an ancillary measurement, and not part of AERTRACC, but Figure S1 shows a lot of high-time resolution, high contrast data that could be certainly utilized to assess what ideally AERTRACC can achieve. So I would encourage the authors to add the PMF factors to Figure 5 (and Tables S3-S5), and the sampling periods to Figure S1, which should make current trends clearer, at the very least. In that context (assuming the AMS is a regular instrument on MoLA), the authors could also consider adding some discussion of to what extent the ancillary variables used in the last section could be augmented by using the f43, f55, f57, f60 from the AMS real time feed.

**Answer:**

We agree that we underutilized the PMF analysis and therefore added the PMF factors to Figure 5 and Tables S3-S5, as well as the sampling periods to Figure S2 (previously S1). In addition, we included further hypothetical sampling scenarios based on sampling conditions using AMS data as suggested and also added them to Figure 5. These additional sampling scenarios are based on the fractional contribution of individual marker ions to the total AMS organics signal. We used f60 as a marker for biomass burning OA and f55 as well as the ratio f55/f57 as markers for cooking-related OA.

As shown in Figure 5, BBOA as well as COA show strongly enhanced concentrations in the sourcerelated sampling intervals, independent on the definition of the sampling scenarios, while OOA concentrations are very similar in the source-related and the background sampling intervals. When the sampling scenarios were defined on the basis of AMS marker ions, an effective enhancement of the respective source-to-background concentration ratio was observed for several of the measured variables. This enhancement is especially large for those variables, which are directly associated with the applied marker ions, used for definition of the sampling scenarios, e.g. enhanced BBOA source-tobackground ratio when f60 is included in the sampling scenario definition. We added a discussion of these additional results to Section 4.3.3, which also includes a discussion on the limitations of this approach, due to the relatively low time resolution of the AMS measurements with typically 15-60 s, which does not allow the reliable sampling of the source aerosol under quickly fluctuating wind conditions.

"The combination of PNC measured by CPC and wind direction was evaluated as additional sampling scenario (Wind+CPC). Further sampling conditions were defined based on the AMS data using fractions of the organic signals at single m/z, e.g. at m/z 55 as  $f_{55}$ , to test whether a potential use of the AMS for AERTRACC control could improve aerosol separation. The selection of a combination of wind direction and  $f_{55}$  (Wind +  $f_{55}$ ) as well as  $f_{55}$  and the ratio  $f_{55}/f_{57}$  (Wind +  $f_{55} + f_{55}/f_{57}$ ) was based on known markers for COA while the combination of wind direction and  $f_{60}$  (Wind +  $f_{60}$ ) was based on the known marker for BBOA. The limit values in the sampling condition definitions were chosen from literature values for these aerosol types (Elser et al., 2016; Mohr et al., 2009; Mohr et al., 2012; Saarikoski et al., 2020)."

**and**

"The mass concentrations of black carbon (BC), polyaromatic hydrocarbons (PAH), organics measured by AMS, the AMS PMF factors BBOA, COA, and OOA, PM1 as well as PNC measured by CPC and OPC were used to compare how well different sampling scenarios separate between source emissions and background."

**and**

"The largest ratios for almost all variables besides the AMS-based ones, and consequently the most effective separation of source-related and background aerosol, were achieved when elevated PNC measured by OPC additional to the right wind direction were used as sampling condition (Wind+OPC)."

**and**

"Inclusion of the AMS data in the AERTRACC control using the fractional signal intensity of known marker m/z could improve specific sampling for certain aerosol types. For COA, higher source/background ratios were achieved with the Wind +  $f_{55}$  sampling scenario, compared to the other scenarios, and even higher ones with the Wind +  $f_{55}$  +  $f_{55}/f_{57}$  scenario as it is more specific for COA. Regarding the potential sampling times especially within the latter scenario, the times are quite limited due to the very specific conditions and possibly due to shorter COA emission periods compared to the more dominant BBOA. The Wind +  $f_{60}$  scenario enables the most effective separation for BBOA combined with potential sampling times comparable to the Wind+OPC scenario."

**and**

"Therefore, for scenarios like Wind stable and Wind +  $f_{55}$  +  $f_{55}/f_{57}$ , longer overall measurement periods in the vicinity of the source are necessary to reach sufficient sampled aerosol mass."

In the summary, a sentence regarding a potential addition of the AMS for AERTRACC control and its benefits was added.

"A future addition of AMS for AERTRACC control would offer the possibility to define specific sampling conditions for certain aerosol types, like BBOA and COA, derived from AMS measurements, using known markers."

**Minor comments:**

The manuscript often uses "aerosol" when it really means "aerosol type", suggest rephrasing.

**Answer:**

We agree with the reviewer and revised the whole manuscript accordingly.

L29: "Within multiphase processes aerosol interacts with atmospheric gases forming new substances". While this is certainly true, in many (most) cases the chemistry happens in the gas phase and the products end up in the particle phase by simple partitioning, so consider rephrasing.

**Answer:**

Thank you for the suggestion. We rephrased the sentence. (L1).

"Various chemical and physical processes lead to permanent changes of the aerosol properties, like the particle size and composition."

L36: Seasalt (as well as dust) is by far the largest primary aerosol type by mass, consider revising.

**Answer:**

We rephrased the sentence (L35).

"Primary particles can be related to anthropogenic sources like combustion processes of fossil fuel and biomass as well as natural sources emitting e.g. sea salt and dust."

It's a minor point, but I find the distinction between the FIGAREO-CIMS and the "TD-HRTOF- CIMS" not helpful and mostly distracting. It's the same instrument, and per the AERTRACC description the authors are even using the FIGAREO inlet for their filter desorption. So it is the same instrument, and the authors are just using it in offline mode, no need to suggest that there is a difference beyond that.

**Answer:**

Thank you for pointing us to this potential source of confusion. We used "TD-HRTOF-CIMS" as name for the analysis method not the instrument which is used for analyzing the filters and TDTs. As instrument, we used the HR-TOF-CIMS from Aerodyne with the FIGAERO inlet for the filters and a custom build inlet for the TDTs. We avoided using the term "FIGAERO-CIMS" because for the TDT analysis we used a different inlet.

To clarify the difference between method and instrument, we revised the manuscript and stated "TD-HR-ToF-CIMS" explicitly as analysis method in the abstract, introduction, and Section 2.5:

"Information on chemical compounds in the sampled aerosol is accomplished by thermal desorption chemical ionization mass spectrometry (TD-CIMS) as analysis method."

**and**

"TD-HR-ToF-CIMS (thermal desorption high resolution time-of-flight chemical-ionization mass spectrometry) was chosen as analysis method offering high resolution mass spectra combined with high sensitivity and low sample fragmentation as well as minimized sample preparation effort (Aljawhary et al., 2013; Mercier et al., 2012; Yatavelli et al., 2012)."

**and**

"For analysis of the samples for this study the TD-HR-ToF-CIMS method was used with the HR-ToF-CIMS (Aerodyne Research Inc., USA) coupled to the FIGAERO inlet for filters and a custom-built inlet for TDT."

L105: "non-refractory chemical composition of submicron particles". This sounds as if the AMS is part of the standard MoLA suite, but that's not what the Table reflects. Please make it consistent.

**Answer:**

Thank you for the hint. Actually, the AMS is part of the standard MoLa suite, however it was not used so far for controlling the AERTRACC sampler due to technical reasons. We included the AMS in Table 1 and, based on your last general comment, we added to the footnote that the AMS data are currently not used for AERTRACC control but might be implemented in the future.

Just a general comment: it seems that the flows are simply checked before startup and then left to their own devices. Given that a typical sample day is probably 4-6 h before the TDTs and filter holders are taken out, relying on the needle valves alone seems fine, but there could still be some variations that could potentially lead to turbulence while valve switching. So some monitoring equipment might be wise...

**Answer:**

Thank you for this suggestion. Indeed, it would be a valuable addition to AERTRACC to avoid any drifts and minimize uncertainties of the sample flows and to simplify flow settings by using mass flow controllers instead of needle valves. For cost reasons this was not implemented yet, but will be considered for future studies. We added to the manuscript in Section 2.2 that the flow rate did not change during control measurements and that mass flow controllers would avoid any fluctuations of the flows.

"No change of flow rates was observed during test measurements. Replacing the needle valves by mass flow controllers for future studies is planned to ensure constant flow rates and to simplify flow settings." Section 2.3: It should be stated in what language the software was developed (Igor Pro, I assume) and also what type of license is used for it.

**Answer:**

Thank you for the remark. It is correct, we used Igor and added this information to the text (Sect. 2.3).

"It was developed with Igor Pro (Version 6.3, WaveMetrics, Inc., USA)."

Section S1: What organic density is used? 1.4 g/cc or the variable one based on Kuwata et al (2012). For source studies the latter seems like the better choice

**Answer:**

We agree that the determination of organic density based on Kuwata et al (2012) is the better choice and used it for our study. We added this information to the SI together with a reference for further details on OPC data treatment.

"The PM1 mass concentrations were calculated from the combined particle number size distributions of FMPS (dp = 5.6 - 560 nm) and OPC ( $dp = 0.25 - 32 \mu$ m) assuming spherical particles with a density calculated using the AMS and black carbon data based on the equation of Kuwata et al. (2012) for organic density and Salcedo et al. (2006) for overall density."

**and**

"Details on OPC data treatment like the conversion from optical diameter to geometric diameter are provided in Drewnick et al. (2020)."

---

## Author Comment (AC2)

Dear reviewer,

we thank you for your valuable feedback to our manuscript, which we believe makes it more valuable to the readers. Within the letter, we commented on and replied to all suggestions and requests. The changes made to the manuscript are written italicized in quotation marks within the reply. Within the manuscript, we marked all changes with track changes.

**Reply on "Comment on amt-2022-206" of Anonymous Referee #2**

This manuscript by Pikmann et al., describes a novel automated batch sampler for aerosols and trace gases integrated into a mobile laboratory platform. The intended use of this instrument is to predefine continuously monitored trace gas measurements, aerosol measurements, and metrological data to control the sampling scheme for targeted airmasses (i.e. urban, biomass burning, cooking emissions, etc). The authors describe the hardware, software and a single source experiment to demonstrate the operation and data product this instrument can provide.

The AERTRACC system is a novel addition to the batch sampling measurement system community. This 'smart' sampler, allows for unattended sampling of different airmasses, without the input of an expert user. The authors describe the implementation of this system clearly and I believe this is well in the scope and interest of the AMT audience. However, I have concerns on the clarity of how this work is presented and the validity of assumptions made for this analysis:

The filters were analyzed with lodide FIAGAREO-CIMS and the thermal desorption tubes (TDTs) were analyzed with a custom desorption unit coupled to the lodide CIMS, however little information is given on analyte sensitivities used in this analysis. It is well known that lodide CIMS sensitivities vary many orders of magnitude by analyte (lyer et al., 2016; Lee et al., 2014; Bi et al., 2021) and are a function of Iodide:H2O ratio, ion optics tuning, and ion molecule reactor (IMR) temperature (Lee et al., 2014; Lopez-Hilfiker et al., 2016; Robinson et al., 2022). Nowhere in this manuscript or SI can the reader find information on what sensitivities are applied to the detected molecular ions in the mass spectra. If the authors did not calibrate, at minimum the assumed sensitivities and dependencies to water and IMR temperature should be described or assumptions stated and the impact to uncertainty. For example, the IMR temperature dependence of sensitivity may explain the large uncertainties found for the TDT calibration samples (62%). I recommend adding sensitivities used for identified compounds from filter analysis in Table 2 and for TDT analysis in Table 3. Additional information on the temperature programed ramp for the FIGAERO-CIMS analysis should also be included (including the IMR temperature time series). Even if sensitivities are not applied to these data, it should be made clear to the reader that reporting ion signal intensities can falsely report molecular abundance in a sample due to the widely varying sensitivities.

**Answer:**

Thank you for pointing us to this issue. We agree that lodide CIMS sensitivities can vary strongly, dependent on the analyte and other parameters. In this manuscript, that deals with the design, characterization, and validation of the AERTRACC sampler, the CIMS is only used to show the enrichment of source-related compounds in the source samples in comparison to those in the background samples. Since we did not intend to determine absolute concentrations of individual compounds in this experiment, no calibrations were performed nor assumptions were made regarding

the sensitivity of the compounds. We assume that ion intensities observed during the analysis of source and background samples, respectively, were directly comparable, as the analysis settings were identical for both sample types. We made this clearer by adding the following statement to Section 4.2:

"Signal intensities for individual compounds were determined semi-quantitatively as a calibration for individual compounds was not feasible."

As the carrier gas for thermal desorption was dried (Section 2.5) and water from the sampling media is desorbed before the organic species we do not expect a strong influence of water onto the analysis.

The temperature ramp for the CIMS analysis is already provided in Section 2.5. Based on your request, we added information regarding the ion optics tuning and the IMR temperature in Section 2.5:

"Tuning of the ion optics was performed before the analysis with formic acid and triiodide for signal intensity, mass resolution, and peak shape using the software Thuner (Tofwerk AG, Switzerland). The IMR conditions were kept constant at 130 mbar and 60 °C."

While the discussion of sampling time delay (3.2) is interesting, I believe the magnitude of the correction to be minimal, and the discussion detracts for the clarity of the methods. I believe most of this section could be moved to the SI.

**Answer:**

The relevance of the time delay is dependent on the typical length of sampling intervals. During our validation experiment, typical sampling periods were in the order of 2-10 s and thus in the same order of magnitude like the time delays. Therefore, knowing the exact time delay is crucial as otherwise due to sampling significant amounts of "wrong" aerosol on the source and background filters/TDTs, the separation of different aerosol types cannot be achieved anymore.

To highlight the importance of the time delay, we added a short paragraph at the end of Section 3.2

"For comparison, sampling periods during the in-field validation (see Section 4) were in the order of 2-10 s. Especially under such conditions, where the sampling periods are in the same order of magnitude as the time delays, it is crucial to consider the time delays for sampling. Otherwise, a significant fraction of the aerosol which does not fulfil the various sampling criteria would nevertheless be sampled and the separation of different aerosol types would not be given anymore."

On a similar note as above, I found the introduction of the AMS PMF analysis to be abrupt and confusing. I believe a more thorough description of the AMS would be useful to introducing these data. It is not clear to the reader if the AMS is a part of the MoLa or was deployed at a different location. I recommend a separate section describing the AMS measurements to make this clear.

**Answer:**

Thank you for this comment. To make the introduction of the PMF analysis less abrupt, we added a short introduction at the beginning of Section 4.3.1:

"During the field measurement period the AMS provided quantitative data on chemical composition of the non-refractory sub-micron particle fraction. For in-depth analysis of the organic fraction, a PMF analysis was performed for source apportionment. The identified aerosol types were biomass burning organic aerosol (BBOA), cooking organic aerosol (COA) and oxygenated organic aerosol (OOA)."

In addition, we added the AMS to Table 1 showing that it is regular part of the MoLa instrumentation. Also, based on comments of reviewer #1, we focus in our manuscript on the description of the AERTRACC sampler and its development and characterization and treat the applied analysis methods rather as auxiliary contributions. Therefore, we did not add in-depth information on AMS and CIMS analysis but rather point the reader to the available literature for these methods.

A discussion of the detection limits for several important compounds (i.e. Levoglucosan, IPN1, IPN2) which appear in large abundance or are close to the Isource/Ibackground = 1 would be helpful to the reader, as it appears the 10% or 62% error bars are applied to all compounds detected (either by filter or TDT analysis). Furthermore, a more rigorous discussion of how these errors are determined would be useful.

**Answer:**

Thank you for mentioning this important point. As stated in our reply to your first comment, we did not determine the sensitivity of individual compounds and only used relative signal intensities for source and background samples. The detection limit, in terms of ion signal intensity, was determined based on the signal intensity variability during times of the filter/TDT analysis process when no sample was desorbed for each ion individually. Ion signal intensities measured during desorption of the samples were all above the detection limit (3\* $\sigma$ (background)) for the individual ions, for most of them more than an order of magnitude above LOD. We added this information to Section 4.2.

"Independently of the sampling media, the ion signal intensities during desorption of the samples exceeded the limit of detection (three times the standard deviation of the molecular background) for all reported samples and ions, with the majority of samples and ions showing an excess by at least an order of magnitude."

Concerning the determination of the errors we added more information in Section S4 how these errors were determined and in Section 2.5 a reference to Section S4 was added.

"To determine the reproducibility, several samples were prepared with equal sample amounts by simultaneously sampling the same aerosol onto multiple filters and TDTs. For the overall reproducibility, the standard deviation over all samples for all individual compounds, which were identified in this study (Section 4), was calculated and then these standard deviations were averaged over all compounds.

As error for the signal intensity of individual compounds the uncertainty, derived from the reproducibility determination, and the error from a Gaussian error propagation of the standard deviation of the blanks and the samples were compared and the larger one was chosen. The signal intensity from compounds found on blank filters was negligible in contrast to source and background samples. The error for the ratios was calculated using Gaussian error propagation from the errors of signal intensity of source and background samples. Error bars of the overall source ratios represent the standard error of the ratios of all ions assigned to the respective sources."

Do the sample media (filters and TDTs) have to be manually changed and at what frequency? It is unclear to the reader if this system can sample for several hours/days unattended or only short timescales (hours/minutes).

**Answer:**

Thank you for pointing that out. Yes, the sample media have to be changed manually.

There is no general upper time limit for unattended sampling. However, the maximum filter load for the selected analysis method (to avoid contamination of the instrument) and the breakthrough volume of the TDTs (to avoid loss of sample material), together with the selected sampling flow rates, constitute an upper sampling time limit. These limits are mentioned in Section 2.4.

We added to Section 2.4 the information on manual change for clarification.

"These limits can be included as sampling conditions to stop sampling automatically when the limits are reached. Afterwards the sampling media need to be changed manually. In our experiment, sampling media were changed after typically 1-1.5 h."

Why does AERTRACC not measure sample flows in real time with MFCs or use a more robust flow control system than needle valves such as critical flow orifices in the sample paths? It seems as though flow drift could be a significant uncertainty in this measurement.

**Answer:**

Thank you for this suggestion. Indeed, it would be a valuable addition to AERTRACC to avoid any drifts and to minimize uncertainties of the sample flows and to simplify flow settings by using mass flow controllers instead of needle valves. For cost reasons this was not implemented yet, but will be considered for future studies. We added to the manuscript in Section 2.2 that the flow rate did not change during control measurements and that mass flow controllers would avoid any fluctuations of the flows.

"No change of flow rates was observed during test measurements. Replacing the needle valves by mass flow controllers for future studies to ensure constant flow rates and to simplify flow settings is planned."

Adding the exact masses, chemical formula, and sensitivities used in this analysis to Table 2 and Table 3 would be useful, rather than putting most of that information in the SI.

**Answer:**

As mentioned above, the sensitivities for the substances have not been determined and are therefore unfortunately not available. The chemical formula and the exact masses can be derived from the compound names and, in addition, are listed in the SI. To make the link to this information easier, we added this information also to the table captions.

"Table 2: Selected identified compounds, measured as iodide cluster, from filter analysis and acronyms used for Fig. 4a. For further details see Table S1."

and

"Table 3: Selected identified compounds, measured as iodide cluster, from filter analysis and acronyms used for Fig. 4a. For further details see Table S2."

Could the authors not have driven the MoLa setup to an appropriate background airmass sampling location and manually sampled in order to confirm the automated determination of background airmasses?

**Answer:**

In principle, such an approach would be desirable for verification purposes; however, with our available instrumentation it is unfortunately not feasible. Due to the transient nature of background aerosol, especially in the complex urban environment of our sampling location, measured background aerosol before/after the hours-long sampling time downwind the pizza oven would likely not have been representative of the background aerosol prevailing during the measurement time. In order to obtain representative values, rather a parallel measurement upwind of the source would have been needed, which however would require numerous additional instrumentation which was not available to us during the measurement time. Such a verification experiment could be envisioned in the future as part of a dedicated measurement campaign with numerous partners.

A source/site map and MoLa sampling location, with Hysplit trajectories would improve the clarity of how these airmasses were sampled. It could be put in the SI if the authors do not believe it adds clarity to the main text.

**Answer:**

Thank you for this suggestion. We added a map showing the location of the institute within the city and a magnification to show the location of MoLa and the pizza oven during the experiment in the SI as Fig. S1. A wind rose plot indicates the predominant wind direction. We added a reference to the map in the SI in the manuscript Section 4.1:

"A site map with the measurement location with respect to the city and to the micro-environment including a wind rose plot showing the predominant wind direction can be found in the supplementary information (Fig. S1)."

Since local wind direction changed on a few second basis during the field measurement, HYSPLIT trajectories, which are available on an hourly basis only, likely do not reflect the local transport situation well. Therefore, we refrained from additionally adding HYSPLIT trajectories.

**Minor comments:**

The manuscript often begins a sentence with a conjunction or preposition:

Line 51 : "For chemical analysis, this approach..."

Line 64: "To obtain data with high time ..."

Stylistically, these sentences read better without beginning them with a conjunction or preposition.

**Answer:**

Thank you for pointing this out. Wherever feasible, we changed this throughout the paper.

Line 79: "Currently, no instrument offers detailed chemical analysis...", I believe there a many instruments which provide this capability, one of which you are using (FIAGERO-CIMS), EESI, CHARON, VIA, all come to mind, so I believe this sentence needs to be reworded to more accurately describe the work you are presenting (Lopez-Hilfiker et al., 2019).

**Answer:**

Thank you for this comment. We agree and removed the mentioned statement. Instead we added the mentioned instruments as current high time resolution instruments.

"A few instruments with high time resolution in the order of seconds, sufficient for the analysis of transient aerosol occurrences, combined with detailed analysis were developed in recent years, such as the EESI-ToF (electrospray ionization time-of-flight mass spectrometer) (Lopez-Hilfiker et al., 2019; Pagonis et al., 2021) and the CHARON-PTR-MS (chemical analysis of aerosol online proton-transfer-reaction mass spectrometer) (Eichler et al., 2015; Piel et al., 2019)."

Line 84: "Since offline methods or highly species-resolving semi-online methods...", this should be reworded, as written it is confusing.

**Answer:**

As suggested, we reworded the sentence.

"Offline and semi-online methods offering highly resolved speciation data do not provide the required temporal resolution and quick online methods do not provide in-depth chemical analysis capability. Therefore, we developed the AERosol and TRACe gas Collector (AERTRACC), which combines the advantages of both approaches."

Line 110: "Table 1...", The AMS should be included in this table.

**Answer:**

We included the AMS in Table 1, however, for technical reasons the AMS data are not yet used for AERTRACC control. Therefore, we added a statement to the footnote that the AMS data are currently not used for AERTRACC control but might be implemented in the future.

Line 135: "Downstream of the cyclone..."

**Answer:**

Thank you for the hint. We rephrased this sentence.

"Inside MoLa the inlet tube is split into two main paths, which are both split again, in total into four sampling paths. Main path 1 (see Fig. 1b) contains a PM1 cyclone (URG, USA, flow rate 16.7 L min-1). Downstream the cyclone, main path 1 and is connected with to main path 2 with a cross tube downstream of the PM1 cyclone."

Line 175: "The two available sampling modes for AERTRACC are either all four sampling paths collecting PM1 aerosol or two...", I believe this has been stated clearly already and is a somewhat redundant sentence.

**Answer:**

Thank you for the remark. We removed the redundant part and combined this sentence with the following one.

"Two sampling modes are available,  $PM_1$  and  $PM_1+PM_{10}$ . For the  $PM_1+PM_{10}$  sampling mode, the same sampling conditions are used for each  $PM_1 / PM_{10}$  sampling path pair."

Line 182: "...EDM", has this been defined somewhere already?

**Answer:**

The acronym EDM is defined in Table 1, in which the MoLa instruments are listed.

Line 226: "...CIMS", has this been defined already?

**Answer:**

CIMS is defined in the introduction.

Line 364: "..is only based on a literature review." The literature review methodology referenced here is not clear to the reader. Please provide references and more detail in the SI.

**Answer:**

Thank you for this comment. Apparently, the word "review" is misleading here, as we did not attempt a complete literature survey. We rather searched through the literature whether the individual substances of interest were previously reported with relation to a specific aerosol type.

To clarify, we exchanged "review" with "search" in Section 4.3.2 and mentioned the criteria for identification of the substances in Sect. 4.2:

"The molecular formula of identified ions was determined for individual peaks; and individual species were identified through the molecular formula, detectability by Iodide-CIMS and occurrence in literature references (further details see Table S1)."

Line 376: "... the ratio is expected to be *on* the order of one."

**Answer:**

Thank you for the remark, we corrected it.

Line 384: "In absolute concentrations...", please provide more information on how absolute concentrations are determined in this work or remove the sentence.

**Answer:**

We removed the sentence, as we did not determine the CIMS sensitivity for measuring levoglucosan.

Line 394: "Some of those species, associated with cooking and biomass burning, can also originate from various other emissions sources and were assigned to the mixed group." Couldn't this also be due to improperly assigning sampling criteria? This assumes the authors perfectly setup the sampling criteria, does it not?

**Answer:**

The reason these species were assigned to the mixed group is because we found in the literature that these species were previously also measured related to other types of emissions apart from cooking and biomass burning. To clarify this, we added that the assignment of this species to the mixed group is based on the literature search.

"Based on a literature search, some of those species, associated with cooking and biomass burning, can also originate from various other emission sources and were therefore assigned to the mixed group."

Line 404: "...have partially ratios on the order of one..."

**Answer:**

Thank you for the remark, we corrected it.

---

## Author Response (AR2)

Dear editor,

we wrote a short reply to the reviewer´s comments to help you to understand our reactions to them. The changes as suggested by the reviewer are written italicized in quotation marks. If we disagreed with the reviewer, we wrote a short reply. Within the manuscript, we marked all changes with track changes.

Major comments:

The authors have certainly made a major effort to address the comments from the first round, and the resulting manuscript is certainly a lot clearer now. I have a few small comments on the response, detailed below. I have also reread the manuscript and I added a few minor comments that mostly address readability. Overall, I think this manuscript will be ready for publication once these comments are addressed.

- The clarification regarding the multiple valve switching nature of the sampler is appreciated. One additional question comes to mind, which is a) what is the minimum sampling interval the software could do (and why?) and b) is that interval (2s per L328?) really desirable/appropriate for ambient sampling?. On the latter point, each valve switch introduces the possibility of either sample contamination (from stuff absorbed on the valve ball) and sample volatilization (from turbulence introduced by the pressure fluctuations). Both of these are admittedly very hard to quantify, but I would be curious what the authors' thoughts are on this subject.

*Section 2.3:*

*"During measurements when air masses containing different aerosol types reach the inlet, the sampler switches automatically between the according sampling paths based on the evaluation of the sampling conditions* . *The evaluation is performed each second based on the highest available time resolution of the instruments, hence the valves can be switched on a 1s-base as well. While frequent switching of the valves introduces frequent flow and pressure disruptions in the sampler, these are not expected to produce enhanced sampling artefacts by e.g. re-volatilization of material from the tube surface or the filters, compared to less frequent switching scenarios. Therefore, switching between different sampling paths typically occurs multiple times within an experiment of hours of duration, which is in contrast to conventional continuous sampling. Although the AERTRACC is primarily designed for stationary measurements, it is also possible to sample during mobile measurements if the air mass segments are large enough to differentiate between them on a few seconds time scale. The flowrate sub-window contains information on the flow setup of the AERTRACC sampler (Fig. 2b). Here, the user enters the flow rates, which are adjusted with the individual needle valves. The graphical user interface automatically provides the combined flow rates at critical devices, such as the inlet cyclone, and thus supports the correct selection of the individual flow rates in order to match their required flow conditions. Furthermore, in this window the MoLa inlet height is entered. This information is used to select the correct delay times between registration*

*of the sampling status, i.e. sampling or non-sampling, and the activation or de-activation of flows through the individual sampling paths (see Sect. 3.2)."*

Reply:

As mentioned in the added text, we do not expect enhanced sampling artefacts due to the frequent switching of the valves. Furthermore, the valves are located downstream the sampling media to avoid any contamination from the valves.

- Regarding the discussion of volatilization of the analyte, I suppose what the authors are trying to say is that at worst it will lead to biased partitioning data (although given the complications in the calibration of gases and aerosols, these are likely not distinguishable in the data from the "true" partitioning). So this could be stated explicitly. Furthermore (and I might be overinterpreting the author's intentions here, this is just my reading of it), typically (e.g. see the recent Tong et al, 2022 paper) it is a given that the results from the molecular technique are not really used for absolute quantification, but for molecular ID'ing. Advanced statistical techniques (e.g. constrained PMF in the Tong et al example) can then be used to further constrain the actual sensitivity/quantification on a molecular PMF basis, bypassing to a large extent the need for single molecule calibrations as requested by Reviewer #2. Up to the authors if they want to make this case explicitly in the paper, but realistically it seems that routine analysis of AERTRACC sampled TD-CIMS data is probably going to be handled in a similar fashion, and that hence both absolute sensitivities and, to a lesser extent, the aforementioned quantification biases can be addressed that way.

*Section 5:*

*Section 4.3.2:*

*"The ratio of the ion signal intensity for selected identified species from the pizza oven and the background samples was calculated for the filter and the TDT samples (Fig. 4), respectively, to show which of the species mainly originate from background and which ones are associated with the source emissions. Additionally, the average ratio for all species assigned to only background (aged/traffic) and oven emissions (biomass burning/cooking – BB/C) as well as both groups (mixed) were calculated for comparison. The assignment to the sources must be regarded as a rather preliminary one, as the apportionment is only based on a literature search. The list of identified species and used acronyms is shown in Table 2 and Table 3 for the filter and TDT samples, respectively. Substances found on the filters and TDTs differ mainly due to gas-particle partitioning and the selectivity of the TDT adsorbents. Volatilization of material from the filters and subsequent sampling in the TDTs could lead to biased information on the partitioning of substances, however, within the uncertainties of the analysis, this effect is presumably not significant."*

*Section 4.2:*

*"Signal intensities for individual compounds were determined semi-quantitatively as a calibration for each compound was not feasible. This allows determination of relative concentrations in separate samples as well as supporting PMF analysis for quantitative determination of aerosol type concentrations (similar to the approach by Tong et al., 2022). Independently of the sampling media, the ion signal intensities during desorption of the samples exceeded the limit of detection (three*

*times the standard deviation of the molecular background) for all reported samples and ions, with the majority of samples and ions showing an excess by at least an order of magnitude."*

- Having said that, I am not clear how you determined that "oligomerization...was not observed in this study". A comparison of the CIMS TD profile with other volatility methods (e.g. bulk TD) would be required for this, and I do not think these data were recorded. It is probably ok to write that "the effect of oligomerization appears minor in our testing", but everything beyond that seems unsupported.

*Section 2.5*

*"The reproducibility of the integrated ion signal intensity of different calibration compounds, determined through laboratory experiments, was found to be 10% for filter and 62% for TDT samples (details see Sect. S4). Oligomerization during analysis with CIMS might occur (Lopez-Hilfiker et al., 2015) but* *appeared minor in our testing."*

- While adding the programing language of the control software is appreciated, I believe it would still be appropriate to mention, either in the paper or in the data availability statement, if the software is publicly available. If so, it would be good to specify under which license and if not, if there are plans to make it available in the future.

Section 2.3:

*"The AERTRACC control software (ACS) is the interface between the MoLa online measurements and the sampling system and is integrated into the MoLa data acquisition software for simple and direct access to the data. It was developed in Igor Pro (Version 6.3, WaveMetrics, Inc., USA) and is available from the authors upon request."*

- L241: I used 10 ug sm3 as a (US-centric) typical number in my review. You could use something more typical of West German urban conditions, and add a reference to support it. Regardless, your subsequent statement covers it.

*"The chemical analysis of aerosol samples (like e.g. when using FIGAERO-CIMS measurements of organic compounds) typically requires sampled mass in the order of 1 μg in order to exceed instrumental detection limits, depending on the specific analysis method. In*  *urban conditions with organic mass concentrations of 5-10 μg m$^{-3}$ (Chen, et. Al, 2022), with a sample flow rate of 7.5 L min-1 and with approximately 10 % of the time sampling source emissions (like in our validation experiment, see Section 4), a total sampling time in the order of*  *1-2 hours would be needed to collect enough material for analysis."*

- L425: DL=3xsigma Background is a conservative DL definition by some practitioners' standards, so overall it does look like all your compounds were actually roughly above DL. The large SD of the TDT, however, is notable. Is this an issue with the much higher concentration of these compounds in the background, or rather was less material collected on the TDTs compared to the filters? As noted by Reviewer, a little bit more detail on how exactly the combined error bars in Figure 4 come about would be helpful.

Reply:

This issue is explained in more detail in the Supplemental Material (Section S4) of the revised version. Apparently, likely because no tracked changes version could be submitted for the SI, the reviewer has missed this information in the revised SI.

- Revised Figure 5: This is very nice, thank you for making it! Please consider adding a vertical grid, the plot is pretty busy and this might improve readability. I would also note that the S/B for the AMS cases might improve if you ever chose to operate the AMS at 0.5 Hz or something like that.

[Figure]

"Inclusion of the AMS data in the AERTRACC control using the fractional signal intensity of known marker m/z could improve specific sampling for certain aerosol types. _This is especially the case if the AMS is operated with shorter averaging intervals to capture short-time variations of air masses containing different aerosol types_. For COA, higher source/background ratios were achieved with the Wind + f55 sampling scenario, compared to the other scenarios, and even higher ones with the Wind + f55 + f55/f57 scenario as it is more specific for COA. Regarding the potential sampling times especially within the latter scenario, the times are quite limited due to the very specific conditions and possibly due to shorter COA emission periods compared to the more dominant BBOA. The Wind +

*f60 scenario enables the most effective separation for BBOA combined with potential sampling times comparable to the Wind+OPC scenario."*

- While again exact quantification is not the goal, in my view, I wonder, also in the context of the PM1/PM10 comparisons, if adding a gravimetric filter analysis of both denuders and filters as a routine step prior to the thermal desorption would be a helpful addition for general QC of the data.

*Section 4.3.2:*

*"In conclusion, for the filter samples the chosen sampling conditions for the background and source emissions proved to be suitable to sample the source emissions separately while the background emissions are found in approximately equal concentrations on the source and background filters at least based on the identified compounds.* *A gravimetric analysis of the samples could be performed in addition to the chemical analysis to extend the general information on the sampled aerosols.* *For the TDT samples the shown ratios indicate a weaker separation of source and background emissions, likely because most of the identified compounds can originate from both, background and source emissions, and no distinct markers were found for the source emissions. "*

- The SI does not seem to have been revised; hence, I have not reviewed it.

Minor comments:

L21: Replace "(PM1 and PM10)" with "(with PM1 and PM10 cutoffs, respectively)"

*"Particle and gas phase of each aerosol type, e.g. source emissions and background, are sampled onto separate filters with $PM_1$ and $PM_{10}$ cutoffs, respectively, and thermal desorption tubes, respectively."*

L30: is "permanent" necessary here?

*"Various chemical and physical processes lead to  changes of the aerosol properties, like the particle size and composition."*

L42: Consider "are generally classified into" instead of "consist"

*"Atmospheric aerosol are generally classified into  two major chemical fractions, the inorganic one with substances like ammonium, nitrate, sulfate, metal oxides, mineral dust, and sea salt , while the organic aerosol, the other fraction, constitutes the more complex part (Fuzzi et al., 2015)."*

L45: Citation needed. Nault 2021 or Southerland 2022 could be used, there are others.

*"Especially fine particulate matter, which has a relevant effect on climate and health, contains usually a large organic fraction (Zheng et al., 2020). These particles consist of many individual components but only a small fraction of them are identified 45 by state-of-the-art instruments (Fuzzi et al., 2015; Johnston und Kerecman, 2019; Zhou et al., 2020)."*

Reply:

The Fuzzi et al., 2015 citation was placed too early in the sentence. This review paper is a reference also for the statement about the organic aerosol fraction. We therefore moved it to the end of the sentence (see answer to comment before). For the following sentence a refence was added that supports the statement that organics are an important fraction of fine particulate matter and its negative health effects.

L73: Citation needed for EC/OC sampler

*"A semi-continuous online bulk analysis can be performed with the thermal-optical EC/OC analyzer measuring the hourly concentrations of elemental carbon (EC) and organic carbon (OC) (Zhou et al., 2015)."*

L92: "and quick online methods do not provide in-depth chemical analysis capability". Not sure what "quick" is trying to qualify here, but in terms of "capability" the EESI or CHARON (or VIA) will provide it, I suppose the question is how cumbersome both ops and analysis are... So at a minimum please replace "quick" with "most". The authors could add a half-sentence about offline methods typically providing more detail regardless, and hence being preferred.

*"Offline and semi-online methods offering highly resolved speciation data do not provide the required temporal resolution and  high-time resolution online methods typically do not provide in-depth chemical analysis capability."*

L106: "probing the emission of a pizza oven"

*"Here, we present the design and characteristics of AERTRACC and demonstrate its capabilities in a field experiment, probing the emissions of a pizza oven in a semi-urban environment."*

L132: Please rephrase: "up to either 10 um (PM10) or". More generally, it would probably read better if you removed this detail up here (could replace with "variable aerosol size cuts") and discussed the size cuts in detail further down. Also, both here and below, it seems that adding "nominal" in front of PM10 would be advisable. I mean, the fact that the plumbing can pass PM10 does still not mean that at say, 60 km/h in turbulent urban BL conditions PM10 sampling can really be achieved...And you write in Section 3.1 (which should probably explicitly be referenced here) that you have calculated significant plumbing losses above 3.5 um. So something like "nominal PM10, in practice PM4-PM5" would seem appropriate.

*Section 2.2:*

*"AERTRACC is designed to sample different aerosol types separately on individual sample carriers. The system is incorporated in MoLa with its own inlet and a flow path designed for minimal particle losses, minimizing non-vertical tubes and bends. With four available sampling paths up to four different aerosol types can be sampled separately. It is possible to sample particles up to 10 µm in aerodynamic diameter (PM10) and up to 1 µm (PM1) with two different size cuts on quartz fiber or PTFE filters as well as volatile compounds onto thermal desorption tubes (TDT) filled with adsorbent material (further details in Sect. 2.4). A control software for the AERTRACC sampler was programmed to accomplish separate sampling of different aerosol types based on the MoLa online data (see Sect. 2.3).*

*A schematic overview and a photograph of the sampling system installed in MoLa are shown in Fig. 1. The AERTRACC sampler has its own inlet line (ID = 48 mm), equipped with a $PM_{10}$ inlet head (Digitel, Switzerland, inlet flow rate 30 L min$^{-1}$) for sampling nominal $PM_{10}$, which is mounted on the roof of MoLa. The inlet is located 0.5 m apart from the MoLa online instrument inlet and their heights are adjusted to each other to assure sampling of the same aerosol."*

Additional reply:

We did not add the suggested statement that the $PM_{10}$ would in practice rather be $PM_4$-$PM_5$ because this is not what our inlet loss calculations suggest. Indeed, for particles larger 3.5 µm in diameter, significant particle losses were found. However, as shown in Figure S5 in the supplement, this does not mean that a majority of particles of this size is lost (it is rather a few percent). Even for 10 µm particles the loss in the tubing is well below 50%.

L285: I assume you "thune" when needed, since for a stationary instrument anything else would seem like overkill, so consider clarifying if you are describing your SOP or just the conditions of the text experiment.

*"Tuning of the ion optics was performed before the first analysis procedure with formic acid and triiodide for signal intensity, mass resolution, and peak shape using the software Thuner (Tofwerk AG, Switzerland)."*

L299: It is unclear how the concentration of individual species would be size-corrected if needed.

Reply:

We assume that, in agreement with the text in L299, the reviewer refers to a correction of the measured mass concentrations, not the particle sizes. Indeed, without knowing the size distribution for individual aerosol types, it would be very hard to correct for large inlet losses. However, due to the design of the inlet system, the transport losses are negligible and therefore no correction of the mass concentrations is needed – as stated in the text.

L375: Replace "occurrence in literature references" with "previous mention in the literature"

*"The molecular formula of identified ions was determined for individual peaks; and individual species were identified through the molecular formula, detectability by Iodide-CIMS and previous mention in the literature occurrence in literature references (further details see Table S1)."*

L376: Consider adding the single calibration compound you are using for scaling in parenthesis

*"Signal intensities for individual compounds were determined semi-quantitatively in terms of detected ions as a calibration for each compounds was not feasible. Independently of the sampling media, the ion signal intensities during desorption of the samples exceeded the limit of detection (three times the standard deviation of the molecular background) for all reported samples and ions, with the majority of samples and ions showing an excess by at least an order of magnitude."*

Reply:

No calibration was performed as stated in the text, hence no compounds were mentioned.

L502: The standard reference for f60 use in the AMS is Cubison et al, 2011, suggest adding it here.

*"The limit values in the sampling condition definitions were chosen from literature values for these aerosol types (Cubison et al., 2011; Elser et al., 2016; Mohr et al., 2009; Mohr et al., 2012; Saarikoski et al., 2012; Sun et al., 2011; Xu et al., 2020)."*

L578: "An in-house developed software package" would be clearer, not obvious who the "self" here is...

*"An in-house developed A self-programmed software is implemented in the MoLa data acquisition software for direct data access."*

L582: Consider specifying the effective size range here again.

*"The inlet and transport system was designed for minimal particle losses with typical estimated mass losses below 1 % for particles in the size-range 35 nm up to 3.5 μm."*

References

Tong, Y., Qi, L., Stefenelli, G., Wang, D. S., Canonaco, F., Baltensperger, U., Prévôt, A. S. H., and Slowik, J. G.: Quantification of primary and secondary organic aerosol sources by combined factor analysis of extractive electrospray ionisation and aerosol mass spectrometer measurements (EESI-TOF and AMS), Atmos. Meas. Tech., 15, 7265–7291, https://doi.org/10.5194/amt-15-7265-2022, 2022

B. A. Nault, D. S. Jo, B. C. McDonald, P. Campuzano-Jost, D. A. Day, W. Hu, J. C. Schroder, J. Allan, D. R. Blake, M. R. Canagaratna, H. Coe, M. M. Coggon, P. F. DeCarlo, G. S. Diskin, R. Dunmore, F. Flocke, A. Fried, J. B. Gilman, G. Gkatzelis, J. F. Hamilton, T. F. Hanisco, P. L. Hayes, D. K. Henze, A. Hodzic, J. Hopkins, M. Hu, L. G. Huey, B. T. Jobson, W. C. Kuster, A. Lewis, M. Li, J. Liao, M. O. Nawaz, I. B. Pollack, J. Peischl, B. Rappenglück, C. E. Reeves, D. Richter, J. M. Roberts, T. B. Ryerson, M. Shao, J. M. Sommers, J. Walega, C. Warneke, P. Weibring, G. M. Wolfe, D. E. Young, B. Yuan, Q. Zhang, J. A. de Gouw, J. L. Jimenez, Secondary organic aerosols from anthropogenic volatile organic compounds contribute substantially to air pollution mortality. Atmos. Chem. Phys. 21, 11201–11224 (2021).

V. A. Southerland, M. Brauer, A. Mohegh, M. S. Hammer, A. van Donkelaar, R. V. Martin, J. S. Apte, S. C. Anenberg, Global urban temporal trends in fine particulate matter (PM2·5) and attributable health burdens: estimates from global datasets. Lancet Planet Health. 6, e139–e146 (2022).

M. J. Cubison, a. M. Ortega, P. L. Hayes, D. K. Farmer, D. Day, M. J. Lechner, W. H. Brune, E. Apel, G. S. Diskin, J. a. Fisher, H. E. Fuelberg, A. Hecobian, D. J. Knapp, T. Mikoviny, D. Riemer, G. W. Sachse, W. Sessions, R. J. Weber, a. J. Weinheimer, A. Wisthaler, J. L. Jimenez, Effects of aging on organic aerosol from open biomass burning smoke in aircraft and laboratory studies. Atmos. Chem. Phys. 11, 12049–12064 (2011).